# The host RNA polymerase II C-terminal domain is the anchor for replication of the influenza virus genome

Tim Krischuns [1,4] ✉, Benoît Arragain [2,4], Catherine Isel [1], Sylvain Paisant[1], Matthias Budt [3], Thorsten Wolff [3], Stephen Cusack [2] ✉ & Nadia Naffakh [1] ✉

The current model is that the influenza virus polymerase (FluPol) binds either to host RNA polymerase II (RNAP II) or to the acidic nuclear phosphoprotein 32 (ANP32), which drives its conformation and activity towards transcription or replication of the viral genome, respectively. Here, we provide evidence that the FluPol-RNAP II binding interface, beyond its well-acknowledged function in cap-snatching during transcription initiation, has also a pivotal role in replication of the viral genome. Using a combination of cell-based and in vitro approaches, we show that the RNAP II C-terminal-domain, jointly with ANP32, enhances FluPol replication activity. We observe successive conformational changes to switch from a transcriptase to a replicase conformation in the presence of the bound RNPAII C-terminal domain and propose a model in which the host RNAP II is the anchor for transcription and replication of the viral genome. Our data open new perspectives on the spatial coupling of viral transcription and replication and the coordinated balance between these two activities.

Influenza viruses are major human pathogens which cause annual epidemics (type A, B and C viruses)[1] and have zoonotic and pandemic potential (type A viruses)[2]. The viral RNA dependent RNA polymerase (FluPol) is a key determinant of viral host-range and pathogenicity, and a prime target for antiviral drugs[3]. It is a ~260 kDa heterotrimer composed of the subunits PB1 (polymerase basic protein 1), PB2 (polymerase basic protein 2) and PA (polymerase acidic protein)[4]. In viral particles each of the eight negative-sense RNA genomic segments (vRNAs) is associated with one copy of the viral polymerase and encapsidated with multiple copies of the nucleoprotein (NP) to form viral ribonucleoproteins (vRNPs)[5]. Upon viral infection, incoming vRNPs are imported into the nucleus in which FluPol performs transcription and replication of the viral genome through distinct primed[6] and unprimed[7] RNA synthesis mechanisms, respectively. Determination of high-resolution FluPol structures by X-ray crystallography

and cryogenic electron microscopy (cryo-EM) have revealed that a remarkable conformational flexibility allows FluPol to perform mRNA transcription[8–12]. Recently, first insights into the molecular process of FluPol genome replication were revealed, however the molecular details remain largely to be determined[13–16].

Transcription of negative-sense vRNAs into mRNAs occurs via a process referred to as 'cap-snatching', whereby nascent 5'-capped host RNA polymerase II (RNAP II) transcripts are bound by the PB2 cap-binding domain[9], cleaved by the PA endonuclease domain[8], and used as capped primers to initiate transcription[10]. Polyadenylation is achieved by a non-canonical stuttering mechanism at a 5'-proximal oligo(U) polyadenylation signal present on each vRNA[12,17]. Influenza mRNAs, therefore, harbour 5'-cap and 3'-poly(A) structures characteristic of cellular mRNAs and are competent for translation by the cellular machinery. The accumulation of primary transcripts of

[1]Institut Pasteur, Université Paris Cité, CNRS UMR 3569, RNA Biology of Influenza Virus, Paris, France. [2]European Molecular Biology Laboratory, Grenoble, France. [3]Unit 17 "Influenza and other Respiratory Viruses", Robert Koch Institut, Berlin, Germany. [4]These authors contributed equally: Tim Krischuns, Benoît Arragain. ✉e-mail: tim.krischuns@pasteur.fr; cusack@embl.fr; nadia.naffakh@pasteur.fr

incoming vRNPs leads to de novo synthesis of viral proteins, including FluPol and NP, which are thought to trigger genome replication upon nuclear import[18].

Replication generates exact, full-length complementary RNAs (cRNAs) which in turn serve as templates for the synthesis of vRNAs. Unlike transcription, replication occurs through a primer-independent, de novo initiation mechanism which occurs either at the first nucleotide of the 3′-vRNA template or at nucleotide 4 and 5 of the 3′-cRNA template followed by realignment prior to elongation[7,19,20]. Nascent cRNAs and vRNAs are encapsidated by NP and FluPol to form cRNPs and vRNPs, respectively[21]. Progeny vRNPs can then perform secondary transcription and replication or, late during the infection cycle, be exported from the nucleus to the cytoplasm to be incorporated into new virions.

FluPol associates dynamically with many cellular proteins, among which the interaction with the host RNAP II transcription machinery[22–24] and the nuclear acidic nuclear phosphoprotein 32 (ANP32)[25] are essential for a productive viral infection. A direct interaction between FluPol and the disordered C-terminal domain (CTD) of the largest RNAP II subunit is essential for 'cap-snatching'[26]. In mammals, the CTD consists of 52 repeats of the consensus sequence $Y_1$-$S_2$-$P_3$-$T_4$-$S_5$-$P_6$-$S_7$[27]. To perform 'cap-snatching', FluPol binds specifically to the CTD when phosphorylated on serine 5 residues (pS5 CTD), which is a hallmark of RNAP II in a paused elongation state[28]. Structural studies revealed bipartite FluPol CTD-binding sites for influenza type A, B and C viruses[22,23,29]. For type A viruses, both CTD-binding sites, denoted site 1A and site 2A, are on the PA-C-terminal domain (PA-Cter) and consist of highly conserved basic residues that directly interact with pS5 moieties of the CTD[23].

The host ANP32 proteins consist of a structured N-terminal leucine-rich repeat (LRR) domain and a disordered C-terminal low complexity acidic region (LCAR) and are essential for influenza virus genome replication[30,31]. Differences in avian and mammalian ANP32 proteins represent a major driver of FluPol adaptation upon zoonotic infections of mammalian species with avian influenza viruses[25,31,32]. A type C virus FluPol (Flu$_C$Pol) co-structure with ANP32A shows that the ANP32A LRR domain binds to an asymmetrical influenza polymerase dimer, which is presumed to represent the FluPol replication complex[13]. The two FluPol conformations observed in this dimer would represent the RNA-bound replicase (FluPol$_{(R)}$), which synthesises de novo genomic RNA and the encapsidase (FluPol$_{(E)}$), which binds the outgoing replication product and nucleates formation of the progeny RNP[18]. In addition, the ANP32 LCAR domain was shown to interact with NP[15,16], leading to a model in which the ANP32 LCAR domain recruits NP to nascent exiting RNA, thus, in combination with the FluPol$_{(E)}$, ensuring efficient co-replicatory encapsidation of de novo synthesised genomic RNAs.

Structural, biochemical and functional studies to date have led to a model in which the FluPol binds either to the pS5 CTD or to ANP32, driving RNPs towards transcription or replication, respectively[18]. Here, we provide genetic evidence that the FluPol CTD-binding interface is essential not only for transcription but also for replication of the viral genome. We show that the CTD, jointly with ANP32, enhances FluPol replication activity and we demonstrate in structural studies that CTD-binding to the FluPol is consistent with replication activity. We therefore propose a model in which transcription and replication of the influenza virus genome occur in association with the RNAP II CTD thereby allowing an efficient switching between the two activities.

## Results

### The FluPol CTD-binding interface is essential for replication of the viral genome

Previous studies demonstrated that the FluPol PA-Cter domain is crucial for FluPol transcription by mediating the interaction between the FluPol transcriptase (FluPol$_{(T)}$) and the host RNAP II pS5 CTD for 'cap-snatching' through direct protein-protein interaction (Fig. S1A, FluPol$_{(T)}$)[22,23]. Recent cryo-EM structures of Flu$_C$Pol suggest that the PA-Cter domain is also involved in the formation of the FluPol replication complex[13]. As there are no structures published for an equivalent Flu$_A$Pol complex, we used the Flu$_C$Pol structures to generate a model of the Flu$_A$Pol replicase-encapsidase complex (Fig. 1A FluPol$_{(R)}$ and FluPol$_{(E)}$). According to our model, the key CTD-binding residues are surface exposed and potentially competent for CTD-binding in the FluPol$_{(R)}$ (Fig. 1A, left). Strikingly, FluPol$_{(E)}$ CTD-binding residues are in close proximity to the ANP32-binding interface (Fig. 1A, right). We therefore hypothesised that the FluPol CTD-binding interface may have a role beyond the described association with the pS5 CTD in the FluPol$_{(T)}$ conformation namely in the assembly of the FluPol$_{(R)}$-ANP32-FluPol$_{(E)}$ replication complex.

To assess this dual role hypothesis, we examined whether binding of the A/WSN/1933 (WSN) FluPol to human ANP32A (huANP32A) is altered by previously described FluPol CTD-binding mutations (PA K289A, R454A, K635A or R638A)[23]. For this purpose, we made use of cell-based G. princeps split-luciferase protein-protein complementation assays as described before in refs. [33,34]. As expected from our previous work[23], steady-state levels of the wild-type (WT) and mutant PA protein are similar or slightly reduced in the case of the PA K635A mutant (Fig. S1B) and each mutation reduces FluPol binding to the CTD (Fig. 1B, grey bars) as well as FluPol activity as measured in a vRNP reconstitution assay (Fig. 1B, hatched bars). Remarkably, they also reduce FluPol binding to huANP32A (Fig. 1B, light blue bars) as well as chicken ANP32A (chANP32A) (Fig. 1B, dark blue bars). We then tested mutations previously shown to impair viral multiplication more strongly, either double mutations (PA K635A + R638A in site 1A, K289 + R454A in site 2A)[23] or mutations in site 2A (PA S420E, Δ550-loop)[22]. Steady-state levels of the mutant PA proteins remain unchanged (Fig. S1C). The observed reductions in CTD-binding signal are in the same 25–50% range as with the first series of single mutants (Fig. S1D, grey bars), suggesting that disruption of a single site, 1A or 2A, largely preserves the CTD-binding signal through the remaining intact binding site. Nevertheless, stronger reductions in huANP32A- and chANP32A-binding and FluPol activity are observed with the second series of mutants compared to the first (Fig. S1D, blue and hatched bars, compared to Fig. 1B), thus reflecting their stronger impact on viral multiplication[22,23].

We then quantified the accumulation of viral RNA species (NA mRNA, cRNA and vRNA) in the vRNP reconstitution assay by strand-specific RT-qPCR[35]. All tested mutants show decreased mRNA accumulation levels (Fig. 1C, grey bars). Strikingly, the PA R454A, K635A, S420E and Δ550-loop mutants also show decreased cRNA and vRNA levels, indicating that both transcription and replication are affected (Fig. 1C, light and dark blue bars, respectively). The variations in cRNA and vRNA levels mirror the variations in FluPol chANP32-binding signal shown in Fig. 1B and Fig. S1D. Such a correlation is not observed with the FluPol huANP32-binding signal, most likely because in our G. princeps split-luciferase protein-protein complementation assay huANP32A binds FluPol weakly compared to chANP32, in agreement with previous reports[33]. We speculate that in the presence of huANP32 the assay is not discriminant enough to resolve mutant-to-mutant differences. In the presence of the PA K289A and R638A mutants, cRNA and vRNA levels appear unchanged or even increased (Fig. 1C). However, like the R454A and 635A mutants, they show not only a ~80% decrease of mRNA:vRNA ratios (Fig. S1E, grey bars) but also a ~50% decrease of cRNA:vRNA ratios (Fig. S1E, blue bars), indicating an imbalance in the accumulation of replication products compared to the WT FluPol.

To further support the notion that the FluPol CTD-binding interface is essential for replication, we made use of transcription-defective (FluPol$_{(T-)}$, PA D108A)[8] and replication-defective (FluPol$_{(R-)}$, PA K664M)[36] mutants of the WSN FluPol. When expressed alone, FluPol$_{(T-)}$

and FluPol(R-) show low activity, as expected (Fig. 1D). However, when they are both co-expressed, FluPol(T-) and FluPol(R-) trans-complement each other and the signal exceeds the background defined as the sum of the activity of FluPol(T-) and FluPol(R-) alone. This trans-complementation system was used to reveal a defect in transcription or in replication (Fig. 1E, F). Upon co-expression with the FluPol(R-) mutant, trans-complementation is observed for most of the investigated CTD-binding mutants (Fig. 1E), confirming that they are transcription-defective. Similarly, upon co-expression with the

FluPol(T-) mutant, trans-complementation is observed for most of the investigated FluPol CTD-binding mutants (Fig. 1F) demonstrating that a replication defect is introduced when the CTD-binding interface is mutated.

Overall, our findings show that the FluPol CTD-binding interface, beyond its function for FluPol 'cap-snatching', is also involved in the FluPol-ANP32 interaction and is essential for genome replication. They are not limited to the WSN strain, as a dual-effect of the mutations PA K289A, R454A, K635A and R638A on CTD-, huANP32A- and

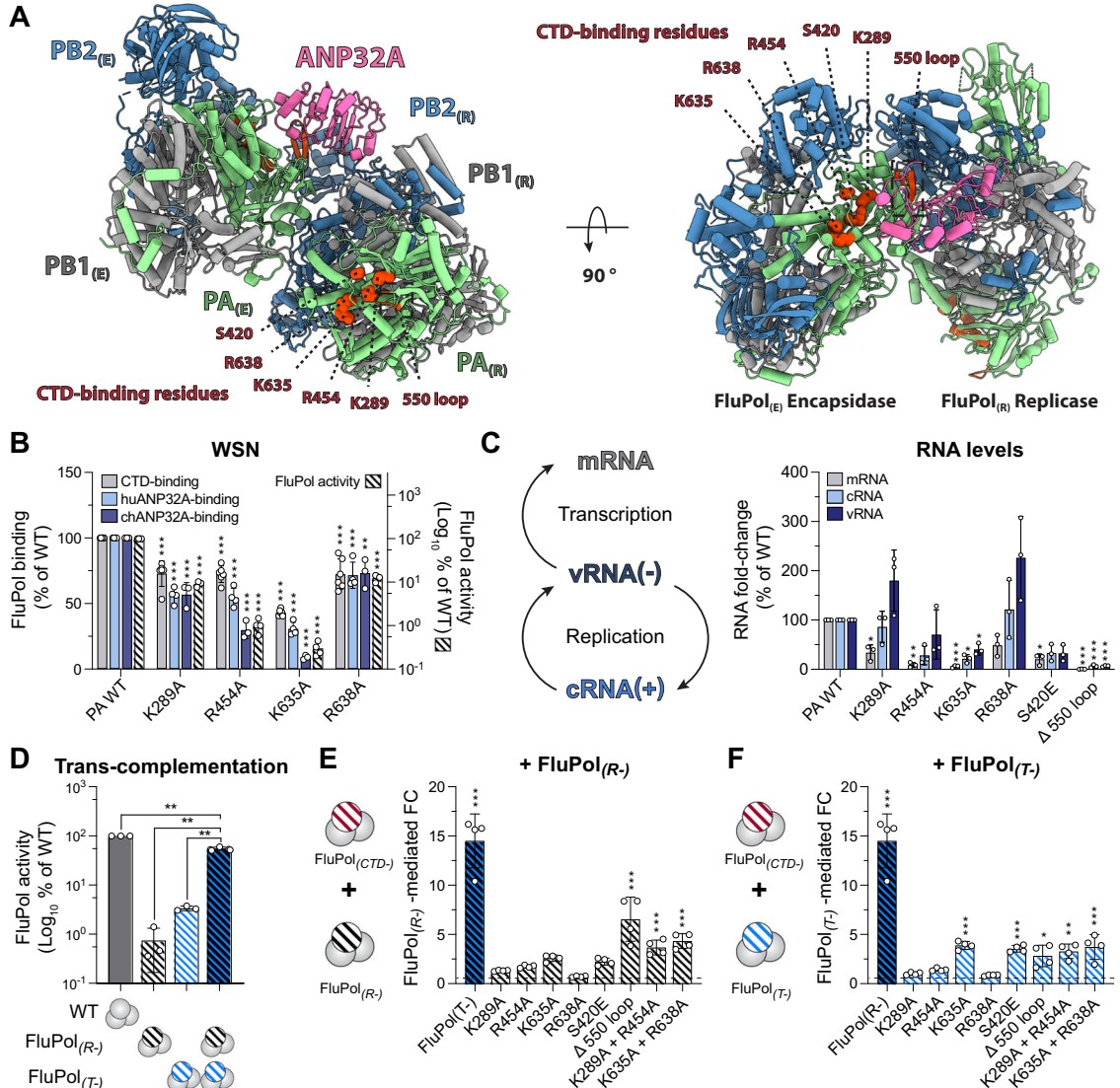

**Fig. 1 | The FluPol CTD-binding interface is essential for replication of the viral genome. A** Model of actively replicating Zhejiang-H7N9 FluAPol replicase-ANP32-encapsidase (FluPol(R)-ANP32A-FluPol(E)). Ribbon diagram representation with PA (green), PB1 (grey), PB2 (blue) and ANP32A (pink). The model was constructed by superposing Zhejiang-H7N9 elongating FluAPol domains (PDB: 7QTL)[43] for the FluPol(R) and apo-dimer Zhejiang-H7N9 FluAPol (PDB: 7ZPL)[43] for the FluPol(E), on those of FluCPol replication complex (PDB: 6XZQ)[13]. ANP32A was left unchanged. Key FluPol CTD-binding residues are highlighted on the FluPol(R) (left) and FluPol(E) (right). **B** WSN FluPol binding and activity assays in HEK-293T cells. Left Y-axis (linear scale): FluPol (PA WT or indicated PA CTD-binding mutants) binding to the CTD (grey bars), huANP32 (light blue bars) and chANP32 (dark blue bars) was assessed using split-luciferase-based complementation assays. Right Y-axis (logarithmic scale): FluPol activity (hatched bars) was assessed by vRNP reconstitution, using a model vRNA encoding the Firefly luciferase (Fluc-vRNA). The luminescence signals are represented as a percentage of PA WT (mean ± SD, $n = 6, 4, 4, 3$, $**p < 0.002, ***p < 0.001$ (one-way ANOVA; Dunnett's multiple comparisons test).

**C** WSN vRNPs were reconstituted in HEK-293T cells using the NA vRNA. Steady-state levels of NA mRNA, cRNA and vRNA were quantified by strand-specific RT-qPCR[35] and are represented as a percentage of PA WT (mean ± SD, $n = 3, *p < 0.033$, $**p < 0.002, ***p < 0.001$, one-way repeated measure ANOVA; Dunnett's multiple comparisons test). **D–F** FluPol trans-complementation assays upon vRNP reconstitution in HEK-293T cells with a model Fluc-vRNA. **D** Trans-complementation of a transcription-defective (FluPol(T-), PA D108A)[8] and replication-defective (FluPol(R-), PA K664M)[36] FluPol. Luminescence is represented as percentage of PA WT (mean ± SD, $n = 3, **p < 0.002$, one-way repeated measure ANOVA; Dunnett's multiple comparisons test). WSN vRNPs were co-expressed with (**E**) a replication-defective (FluPol(R-), PA K664M)[36] or (**F**) a transcription-defective (FluPol(T-), PA D108A)[8] FluPol. Luminescence signals are represented as the fold-change (FC) relative to the background, which is defined as the sum of signals measured when the FluPol(CTD-) and FluPol(R-/T-) were transfected alone (mean ± SD, $n = 4, *p < 0.033$, $**p < 0.002, ***p < 0.001$, two-way ANOVA; Sidak's multiple comparisons test). Source data are provided as a Source data file.

chANP32A-binding is also observed with the FluPol of the human isolate A/Anhui/01/2013 (H7N9) (Anhui-H7N9), a virus of avian origin which harbours the human-adaptation signature PB2 K627 (Fig. S1F, G). Sequence alignments of the PB1, PB2, PA and NP proteins from the influenza A viruses used in this study are shown in Fig. S2.

### Serial passaging of FluPol PA K289A, R454A, K635A, R638A mutant viruses selects for adaptive mutations which restore FluPol binding to the CTD and huANP32A

We previously rescued by reverse genetic procedures recombinant WSN PA mutant IAVs (PA K289A, R454A, K635A, R638A) that were highly attenuated, formed pinhead-sized plaques and had acquired multiple non-synonymous mutations[23]. In order to investigate whether and how the recombinant viruses adapt to defective CTD and ANP32-binding (Fig. 1), we subjected them to eight serial cell culture passages in MDCK-II cells followed by titration and next generation sequencing (NGS) of the viral genome at passage 1 to 4 and 8 (Fig. 2A, p1–4 and p8). The plaque size of the passaged viruses progressively increased (Fig. 2B and S3A), which went along with distinct patterns of adaptive mutations (Source data file), suggesting that cell culture passaging increased the replicative capacity of the PA mutant viruses. True reversions that require at least two nucleotide substitutions did not emerge. The observed second-site mutations are primarily located on the FluPol subunits as well as the NP (Fig. 2C, S3B and S4A) and are mostly surface exposed (Fig. 2D and S4B–D). Several FluPol second-site mutations are located close to the FluPol CTD-binding sites (Fig. 2D, PA D306N, C489R, R490I, R496Q) while some PB2 second-site mutations are located in flexible hinge-subdomains (Fig. 2D, PB2 D253G, S286N, M535I). Overall, the FluPol second-site mutations show no local clustering or common function, indicating that diverse routes of adaptation occurred. NP second-site mutations are also surface exposed, but to our knowledge have not been associated to date with any specific function (Fig. S4D)[37]. The observed reversion pathways most likely represent a fraction of possible reversion pathways, especially as passaging was performed at low MOIs and, therefore, less frequent variants may have been lost.

We first focused on the reversion pathway of PA K289A mutant virus. Recombinant viruses with combinations of FluPol second-site mutations observed along the PA K289A reversion pathway at p1, p4 and p8 recapitulate the progressive increase in plaque size observed during serial passaging (Fig. 2E and S3C), demonstrating that the FluPol second-site mutations are leading to the observed increase in viral replicative capacity. PB2 and PA proteins with second-site mutations which occurred during passaging of PA K289A mutant virus, accumulate at similar levels compared to the WT proteins (Fig. S3D). However, the combinations of FluPol second-site mutations result in sequential increases and decreases of CTD-binding, huANP32A-binding and/or FluPol activity (Fig. 2F). The highest levels of CTD-binding, huANP32A-binding and FluPol activity are observed for the combination of mutations observed at p8 (PA K289A + PB2 D253G + PA C489R), which goes along with the highest replicative capacity observed for the passaged virus at p8 (Fig. 2B, E).

The reversion pathways of the PA R454A, K635A and R638A mutant viruses were analysed in the same way (Fig. S3E). Across the tested combinations of primary FluPol mutations and second-site mutations that occurred during passaging, FluPol activity shows a strong positive correlation with huANP32A-binding (Fig. 2G, left panel) as well as a positive correlation with CTD-binding (Fig. 2G, middle panel) but not with FluPol-binding to other known cellular partners, such as DDX5 (Fig. 2G, right panel) or RED (Fig. S3F)[38].

Overall, we show that serial cell culture passaging of PA K289A, R454A, K635A and R638A mutant viruses selects for adaptive mutations that rescue viral polymerase activity by restoring FluPol binding to the CTD and/or huANP32A, which agrees with our initial observation

that the investigated FluPol mutants are impaired for CTD-binding as well as ANP32-binding (Fig. 1).

### CTD overexpression leads to an increased binding of FluPol to huANP32

To further document a potential interplay between CTD and ANP32-binding to FluPol, we tested the effect of mCherry-CTD (CTD-WT) overexpression on FluPol binding to huANP32A as well as FluPol oligomerisation in cultured cells. A FluPol binding-deficient CTD in which all serine 5 residues were replaced with alanines was used as a specificity control (CTD-S5A)[22]. mCherry-CTD-WT and -CTD-S5A accumulate to similar levels (Fig. S5A), however mCherry-CTD-WT leads to a significant increase of FluPol huANP32A-binding compared to the FluPol binding-deficient CTD-S5A mutant, suggesting that the CTD facilitates FluPol-binding to huANP32A (Fig. 3A). mCherry-CTD-WT overexpression also leads to significantly increased WSN FluPol oligomerisation compared to the FluPol binding-deficient CTD-S5A mutant (Fig. 3B), as measured in a split-luciferase complementation assay[39]. A similar increase in oligomerisation was also observed with the Anhui-H7N9 and B/Memphis/13/2003 FluPols (Fig. S5B). Various oligomeric FluPol species have been described[13,40,41]. There is evidence that the utilised split-luciferase-based assay, when FluPol is overexpressed in the absence of any other overexpressed viral or cellular protein, detects the symmetrical FluPol dimer species[39,41,42]. Upon CTD overexpression, it is unclear which oligomeric FluPol species is affected. However, the oligomerisation signal is still increased in the presence of the PB2 NEQ71-73AAA mutant, which abolishes FluPol symmetrical dimerisation (Fig. S5C–E)[39,41], indicating that CTD overexpression affects FluPol oligomers distinct from the symmetrical FluPol dimer. Moreover, the increased signal upon CTD-WT overexpression is less pronounced in HEK-293T cells depleted for huANP32A and huANP32B (ANP32AB KO) (Fig. S5F, G), most clearly so for B/Memphis/13/2003 FluPol, suggesting that the CTD-dependent increase in FluPol oligomerisation is at least partially dependent on huANP32A and/or huANP32B. Altogether, the data show that CTD overexpression leads to increased binding of FluPol to huANP32A as well as FluPol oligomerisation, suggesting that the CTD can bring two or more FluPols in close proximity with each other in conjunction with ANP32A, thereby promoting FluPol replication.

### De novo FluPol RNA synthesis activity is enhanced in vitro in the presence of CTD and ANP32A

To investigate by biochemical and structural approaches whether in vitro replication is dependent on ANP32A and pS5 CTD, we performed unprimed RNA synthesis assays using recombinant A/Zhejiang/DTID-ZJU01/2013(H7N9) FluPol[43] bound to a 51-mer vRNA loop template (denoted v51_mut_S). This polymerase, referred to below as Zhejiang-H7N9, derives from a human isolate of an avian strain that possesses PB2 E627 together with the mammalian adaptation mutation PB2 N701 (Fig. S2)[44]. The v51_mut_S template sequence derives from joining the first 30 and last 21 nucleotides (nts) of the respective 5′ and 3′ vRNA ends of Zhejiang-H7N9 segment 4. It was modified so that a stalled elongation product of 33 nts could be produced by using only AGC nts or replacing UTP in the reaction by non-hydrolysable UpNHpp (Fig. 3C, light blue letters). This 33-mer cRNA product was estimated to be long enough that its 5′ hook (1–10) could bind to the encapsidase, according to the modelled Zhejiang-H7N9 replicase-encapsidase structure. The addition of all NTPs would in fine produce a full-length 51-mer replication product without poly-adenylation (since the oligo-U sequence is reduced to 4xU due to changing U18 to A)[12], assuming RNA synthesis proceeded to the 5′ end of the template, rather than terminating prior to reading through the 5′ hook (Fig. 3C, dark blue letters).

De novo replication assays were performed with WT Zhejiang-H7N9 FluPol, using three (ACG) or four (ACGU) nts, excess apo-polymerase, that could serve as encapsidase, recombinant huANP32A and

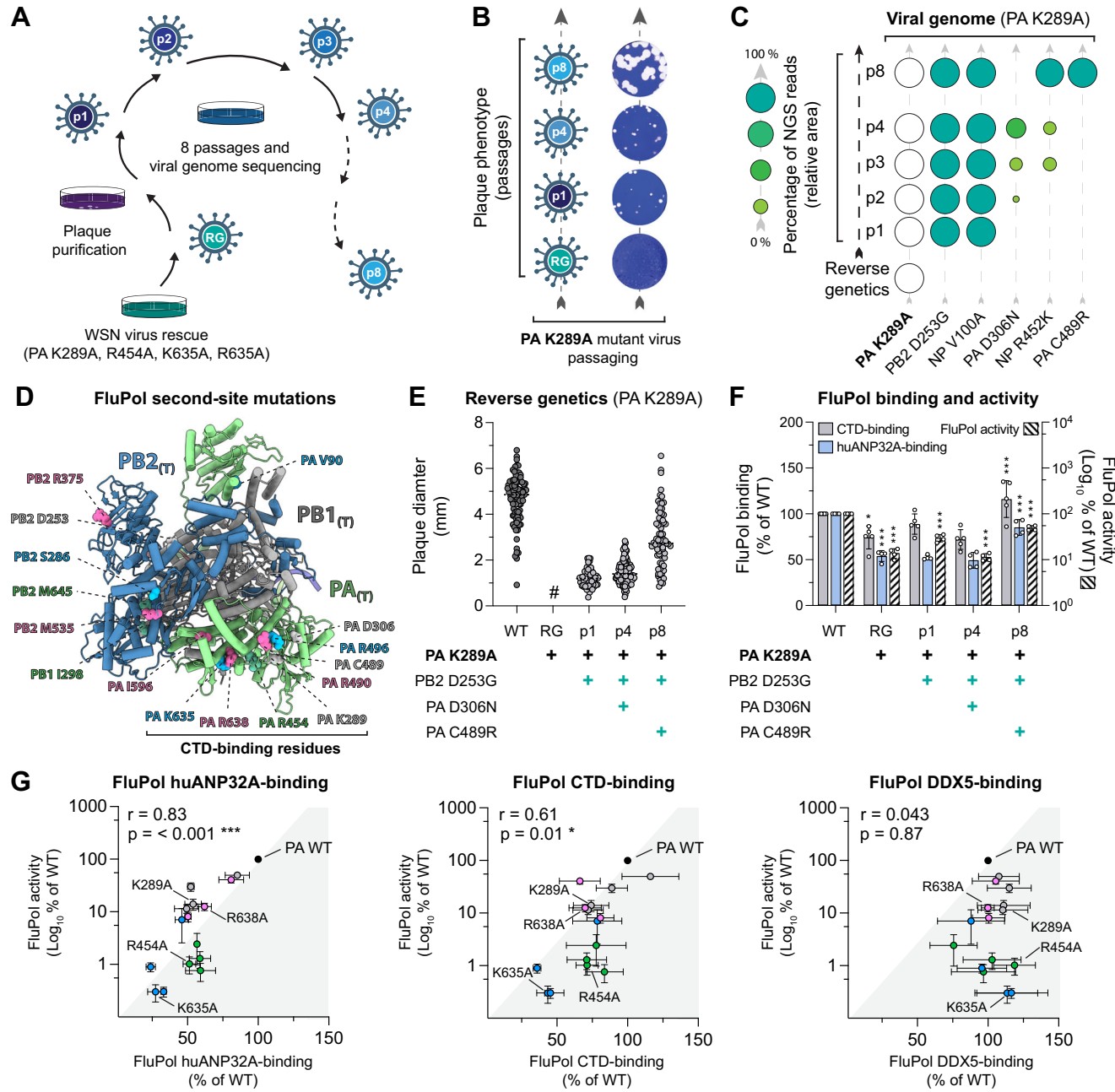

**Fig. 2 | Serial passaging of mutant WSN viruses with mutations at the FluPol-CTD interface selects for adaptive mutations which restore FluPol binding to the CTD and huANP32A.** **A** Schematic representation of the passaging experiment. **B, C** Passaged PA K289A viruses. **B** Plaque phenotype. Representative images of crystal violet stained plaque assays on reverse genetics (RG), p1, p4 and p8 supernatants are shown. **C** Short-read next generation sequencing of the viral genome. Second site mutations found in ≥10% of reads in at least one passage are shown (green circles). The fraction of reads showing a given mutation is indicated by the circle area and shades of green (schematic on the left: <25, 25–50, 50–75 and >75 % of reads). **D** Ribbon diagram representation of FluPol$_{(T)}$ conformation (A/NT/60/1968, PDB: 6RR7)[41]. Second-site mutations observed during passaging of the PA K289A, R454A, K635A and R638A viruses are indicated in grey, green, blue and pink fonts respectively. **E, F** FluPol second-site mutations observed during passaging of the PA K289A virus. **E** Recombinant viruses with PA K289A and the indicated second-site mutations were produced (*n* = 2). Two independent RG supernatants were titrated, plaque diameters (mm) were measured, each dot represents one

plaque (see Fig. S3C). (#) pinhead-sized plaques. **F** Cell-based FluPol binding and activity assays as in Fig. 1B. Left Y-axis (linear scale): FluPol binding to the CTD (grey bars) and huANP32A (blue bars). Right Y-axis (logarithmic scale): FluPol activity (hatched bars). Luminescence signals are represented as a percentage of WT FluPol. Stars indicate statistical significance when passage N is compared to passage N-1, or RG is compared to WT FluPol (mean ± SD, *n* = 5, 4, 4, **p* < 0.033, ***p* < 0.002, ****p* < 0.001, one-way ANOVA; Tukey's multiple comparisons test). **G** For each primary mutation and FluPol genotype observed upon serial passaging, FluPol binding to huANP32A, CTD and DDX5 (x-axis in the left, middle and right panel, respectively) were plotted against the FluPol activity (y-axis). Binding and activity data are represented separately in (**F**) and Fig. S3E. Combinations of mutations which appeared during passaging of the WSN PA K289A, R454A, K635A and R638A mutant viruses are highlighted in grey, green, blue and pink respectively. (mean ± SD, *n* = 4, r: Pearson correlation coefficient, **p* < 0.033, ****p* < 0.001,two-tailed 95% confidence interval). Source data are provided as a Source data file.

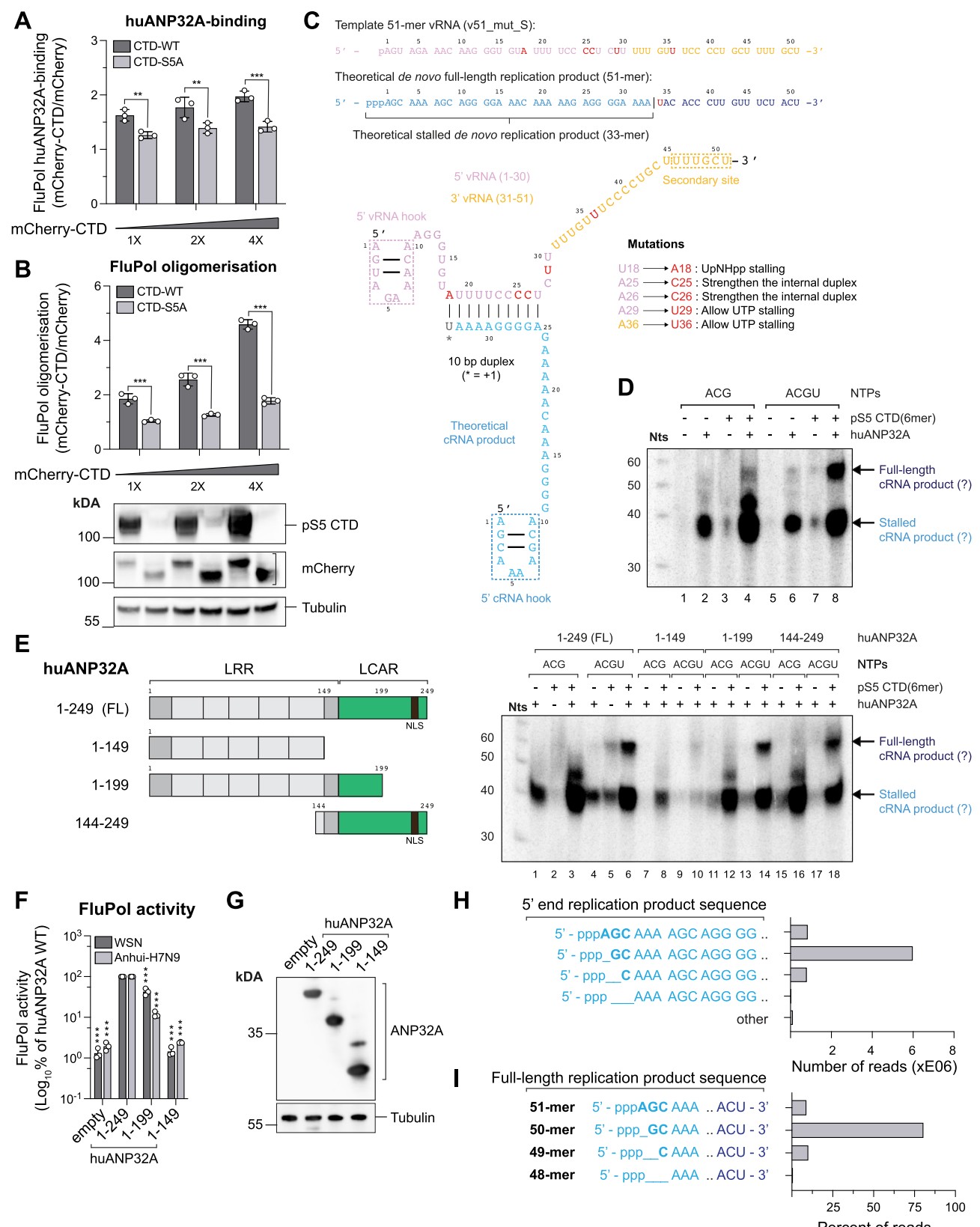

6-repeats pS5 CTD peptide (pS5 CTD(6mer)) (Fig. 3D). In reactions with ACG only (Fig. 3D, lane 1–4), the principal product is around 38–40 nts (with another at ~45 nts), whose significant production is only visible in the presence of huANP32A (Fig. 3D, lane 2) and strongly enhanced when both the CTD and huANP32A are added (Fig. 3D, lane 2 vs 4). When all four nts are added (Fig. 3D, lane 5–8), the presence of CTD

and huANP32A again strongly enhances RNA synthesis with the production of both a similar product as with ACG alone (38–40 nts) and a significant amount of a longer ~55–56 nts product (Fig. 3D, lane 6 vs 8). Although the apparent size of these products does not correspond with the theoretical length of the stalled and full-length products (they are consistently around 5–6 nts higher), nevertheless, we think that

**Fig. 3 | FluPol replication activity is enhanced in the presence of CTD and ANP32A.** WSN FluPol binding to huANP32A (**A**) and WSN FluPol oligomerisation (**B**) in the presence of increasing amounts of pS5 CTD, as assessed using split-luciferase-based complementation assays. Plasmids encoding mCherry or mCherry fused to CTD-WT (dark grey bars) or CTD-S5A (serine five residues replaced with alanines, light grey bars) were co-transfected in increasing amounts (mean ± SD, $n = 3$, **$p < 0.002$, ***$p < 0.001$, two-way ANOVA; Sidak's multiple comparisons test). Luminescence signals are represented as fold-changes compared to untagged mCherry co-expression. In (**B**), cell lysates were analysed by western blot using the indicated antibodies ($n = 1$). **C** 51-mer vRNA template (v51_mut_S) derived from segment 4 of A/Zhejiang/DUID-ZJU01/2013(H7N9)/KJ633805 and used in in vitro replication activity assays. The vRNA 5′end (1–30) and 3′ end (31–51) are coloured in pink and gold, respectively, with introduced mutations in red. The theoretical de novo full-length replication product is coloured in light blue (1–33) and dark blue (35–51), with U34 in red. The expected stalled elongation state is schematically represented. de novo replication activity assays of Zhejiang-H7N9 FluPol using v51_mut_S, in the presence of either 3 NTPs (AUG) or 4 NTPs (AUGC), with or

without pS5 CTD(6mer) and huANP32A full-length (**D**, $n = 3$) or deletion mutants (**E**, $n = 3$). Tentative full-length and stalled replication products are indicated by an arrow. LRR leucine-rich repeat, LCAR low complexity acidic region, NLS nuclear localisation signal. Nts: molecular weight marker. **F** WSN and Anhui-H7N9 vRNPs were reconstituted in ANP32AB KO cells with a model Fluc-vRNA, and co-expressed with the indicated huANP32A proteins (V5-tagged and fused to SV40-NLS). Luminescence signals are represented as a percentage of huANP32A FL. (mean±SD, $n = 3$, ***$p < 0.001$, two-way ANOVA; Sidak's multiple comparisons test). **G** Lysates of ANP32AB KO cells transiently expressing the indicated huANP32A proteins were analysed by western blot using an anti-V5 antibody ($n = 1$). **H, I** RNA-sequencing of Zhejiang-H7N9-4M FluPol de novo replication products in the presence of all NTPs, the pS5 CTD(6mer) and huANP32A. **H** The number of reads is plotted according to the recurrence of the exact 5′ cRNA motifs indicated on the left. Reads that do not encompass these motifs are plotted as 'Other'. **I** The percentages of full-length replication products are plotted according to the 5′ terminal nucleotide indicated on the left. Source data are provided as a Source data file.

they correspond to these products for the following reasons. Firstly, for both the AGC+UpNHpp and AGCU reactions, we have obtained high-resolution cryo-EM structures of stalled elongation states, unambiguously showing products of size 33-mer and 35-mer respectively (32-mer/34-mer if RNA synthesis starts at G2 of the template instead of U1, see below) that we presume to correspond to the bands observed. Secondly, for the AGCU reaction, the difference in size of the two bands is 16–18 nts, which corresponds to the expected difference between the stalled and full-length (hook read-through) products. We are unable to explain the origin of the ~45 nts product with AGC only. Despite the remaining uncertainties in the assignment of the products, these results demonstrate the synergistic effect of CTD and huANP32A in significantly enhancing de novo cRNA synthesis.

We further investigated which domains of huANP32A are essential for the enhancement of de novo RNA synthesis activity in the presence, or not, of pS5 CTD(6mer) peptide. We compared the full-length huANP32A ('FL' lane 1-6) with huANP32A subdomains corresponding to the LRR ('1–149' lane 7-10), the LRR and LCAR N-terminal region ('1-199' lane 11-14), or the LCAR and LRR C-terminal region ('144–249' lane 15–18) (Fig. 3E). While without addition of CTD, only FL huANP32A supported significant replication product synthesis, we show that the 144-199 region of huANP32A is most critical to observe the synergistic CTD- and ANP32-dependent enhancement of replication product synthesis (Fig. 3E, right panel). These in vitro findings are consistent with cell-based assays in which FluPol activity was investigated in the presence of the same huANP32A C-terminal deletion mutants (Fig. 3F, G). While the presence of the CTD alone was shown to enhance FluPol transcriptional activity in vitro[22,29], enhancement of FluPol replicational activity requires both the presence of the CTD and the 144-249 domain of ANP32, suggesting that CTD and the ANP32 LCAR domain are jointly priming FluPol for replication.

Instead of using the WT Zhejiang-H7N9 FluPol, which forms a robust symmetrical dimer in vitro[22] that could compete with the formation of an active asymmetric FluPol$_{(R)}$-FluPol$_{(E)}$ dimer, we subsequently used a Zhejiang-H7N9 FluPol that harbours four mutations (PA E349K, PA R490I, PB1 K577G, and PB2 G74R) and is referred to below as the 4 M mutant. PA E349K, PB1 K577G, and PB2 G74R were shown to disrupt the symmetrical apo FluPol dimer interface[39], while PA R490I appeared during passaging of the PA R638A mutant virus and increases FluPol-ANP32A binding in a cell-based assay (Fig. S3E). Indeed, Zhejiang-H7N9 4 M shows enriched monomeric particles while preserving a similar in vitro activity when compared to the Zhejiang-H7N9 WT FluPol (Fig. S6A, B). Accordingly, in cell-based assays Anhui-H7N9 4 M shows decreased FluPol oligomerisation, increased binding to huANP32A, while showing activity levels comparable to the WT Anhui-H7N9 FluPol as measured in a vRNP reconstitution assay (Fig. S6C).

To determine whether 5′ triphosphorylated full-length replication products were actually synthesised, we performed next-generation sequencing (NGS) of all RNAs present in the reaction mix (Fig. S6B, left panel, lane 8). To simplify the analysis by NGS of this heterogeneous RNA sample, we first degraded non-5′ triphosphorylated RNAs with a 5′-phosphate-dependent exonuclease. This removes the v51_mut_S template, which carries a 5′ monophosphate, and which is perfectly complementary to the expected product and could have formed unwanted double-stranded RNA (Fig. S6D). NGS analysis revealed that the majority of reads (i.e., premature and near full-length products) start with 5′-ppp**G**C... instead of the expected 5′-ppp**A**GC... (Fig. 3H). Despite the presence of many premature termination products (Fig. S6E), around 80% of near full-length products, are 50-mers, lacking the expected 5′ pppA, with few 51-mers (5′-ppp**A**GC...) and 49-mers (5′-ppp**C**...) (Fig. 3I). This suggests that, in vitro, most RNA synthesis products are initiated de novo with a 5′-pppG at the second position of the v51_mut_S template (3′-U**CG**...).

**Structural analysis shows that both replication and transcription are consistent with CTD-binding to the FluPol**

Using cryo-EM, we then sought to determine structures of actively replicating Zhejiang-H7N9 4M FluPol in the presence of huANP32A, pS5 CTD(6mer) peptide mimic and excess apo FluPol, by freezing grids with the de novo reaction stalled with UpNHpp (Fig. S6B, left panel, lane 4). Several high-resolution structures were obtained from a Krios data collection (Table 1, Fig. 4 and S7). These include FluPol in the recently described intermediate structure, denoted PB2-C(I)/ENDO(T), where the cap-domain is swung out (Fig. 4A, left panel, 3.25 Å resolution)[14,45]. A second structure was obtained at 2.9 Å resolution where FluPol with PB2-C mid-link and cap-binding domains take a replicase-like configuration (PB2-C(R)), with the 627-domain remaining flexible (Fig. 4A, middle panel). However, the PA endonuclease is still in an unrotated transcriptase-like conformation (ENDO(T)), thus precluding the usual interaction with the PB2-NLS domain, as previously observed for Flu$_A$Pol H5N1 replicase conformation (Fig. 4A, right panel)[41]. A third structure was obtained at a higher resolution (2.5 Å), in which PB2-C is not visible (Fig. 4B). These three structures show FluPol in the pre-initiation state mode A, with the 3′ end of the v51_mut_S template (3′-UCG...) in the active site and the priming loop fully extended and well ordered (Fig. 4C, D and S8A). The template is positioned with C2 and G3 respectively at the −1 and +1 positions, whilst only the phosphate of U1 is visible (Fig. 4C, D). In addition, a stalled elongation state was obtained at 2.9 resolution in which FluPol encloses a template-product duplex, with incoming UpNHpp at the +1 active site position, and the 3′ vRNA end bound to the secondary site (Fig. 4E and S8B). Finally, from a similar de novo reaction mix but in the

**Table 1 | Cryo-EM structures data collection, refinement and validation statistics**

| Sample | A/H7N9 polymerase with Pol II pS5 CTD mimic bound in site 1 A/2 A | | | |
|---|---|---|---|---|
| Structure | Pre-initiation state in replicase (R)-like conformation PB2-C(R), ENDO(T) | Pre-initiation state core ENDO(T) | Pre-initiation state in intermediate (I)-conformation PB2-C(I), ENDO(T) | Elongation state stalled with UpNHpp ENDO(T) |
| PDB ID | PDB ID 8PMO | PDB ID 8PNP | PDB ID 8R3L | PDB ID 8PNQ |
| EMDB ID | EMD-17755 | EMD-17782 | EMD-18872 | EMD-17783 |
| **Data collection and processing** | | | | |
| Microscope | ThermoFisher Krios TEM | | | |
| Voltage (kV) | 300 | | | |
| Camera | Gatan K3 direct electron detector mounted on a Gatan Bioquantum LS/967 energy filter | | | |
| Magnification | 105000 | | | |
| Nominal defocus range (µm) | −0.8/−2.0 | | | |
| Electron exposure (e−/Å$^2$) | 40 | | | |
| Number of frames collected (no.) | 40 | | | |
| Number of frames processed (no.) | 40 | | | |
| Pixel size (Å) | 0.84 | | | |
| Initial micrographs (no.) | 6000 | | | |
| Final micrographs (no.) | 5883 | | | |
| **Refinement** | | | | |
| Particles per class (no.) | 33395 | 260565 | 17121 | 14878 |
| Map resolution (Å), 0.143 FSC | 2.9 | 2.5 | 3.25 | 2.9 |
| Model resolution (Å), 0.5 FSC | 3.0 | 2.5 | 3.3 | 2.9 |
| Map sharpening B factor (Å$^2$) | −20 | −30 | −20 | −20 |
| Map versus model cross-correlation (CCmask) | 0.8428 | 0.8901 | 0.8179 | 0.8826 |
| **Model composition** | | | | |
| Non-hydrogen atoms | 16673 | 14464 | 17878 | 14917 |
| Protein residues | 1997 | 1716 | 2149 | 1727 |
| Nucleotide residues | 31 | 29 | 31 | 45 |
| Water | - | 43 | - | - |
| Ligands | 1 Mg | - | - | 2 Mg, 1 UpNHpp |
| **B factors (Å$^2$)** | | | | |
| Protein | 74.00 | 67.24 | 89.19 | 70.02 |
| Nucleotide | 72.82 | 60.96 | 90.96 | 83.39 |
| Ligand | 48.38 | - | - | 68.99 |
| Water | - | 33.97 | - | - |
| **RMS deviations** | | | | |
| Bond lengths (Å) | 0.003 | 0.003 | 0.002 | 0.005 |
| Bond angles (°) | 0.475 | 0.499 | 0.478 | 0.465 |
| **Validation** | | | | |
| MolProbity score | 1.53 | 1.35 | 1.53 | 1.40 |
| All-atom clash score | 4.79 | 4.32 | 6.41 | 3.86 |
| Poor rotamers (%) | 2.03 | 1.24 | 0.21 | 2.01 |
| **Ramachandran plot** | | | | |
| Favoured (%) | 97.77 | 97.70 | 97.00 | 98.24 |
| Allowed (%) | 2.23 | 2.24 | 3.00 | 1.76 |
| Outliers (%) | 0.00 | 0.06 | - | 0.00 |

presence of all NTPs (Fig. S6B, left panel, lane 8), we obtained a structure at 3.43 Å resolution of Zhejiang-H7N9 4 M FluPol in a self-stalled pre-termination state (Table 2 and Fig. S9). In this post-transfer structure, with the PPi reaction product visible, template U17 base pairs with A35 of the product cRNA at the +1 active site position, with a maximally taut connection of the template from the hook to the active site, as previously described[12]. This structure shows that even when all NTPs are present, under these conditions self-stalling prior to the hook can occur, as seen in the in vitro activity assays (Fig. 3D, E and S6A, B).

In all structures, the pS5 CTD is observed bound in sites 1A and 2A. Continuous density between the two sites, with 25 connecting residues coming from five CTD repeats, is observed for the pre-initiation state and the elongation structures (Fig. 4F), whereas for the replicase-like structure, the connectivity is lost between both sites. Despite extensive cryo-EM data analysis, huANP32A was never visualised nor the putative replicase-encapsidase dimer.

Altogether, these results show that in vitro, de novo RNA synthesis assays result in a heterogeneous mix of FluPol conformations

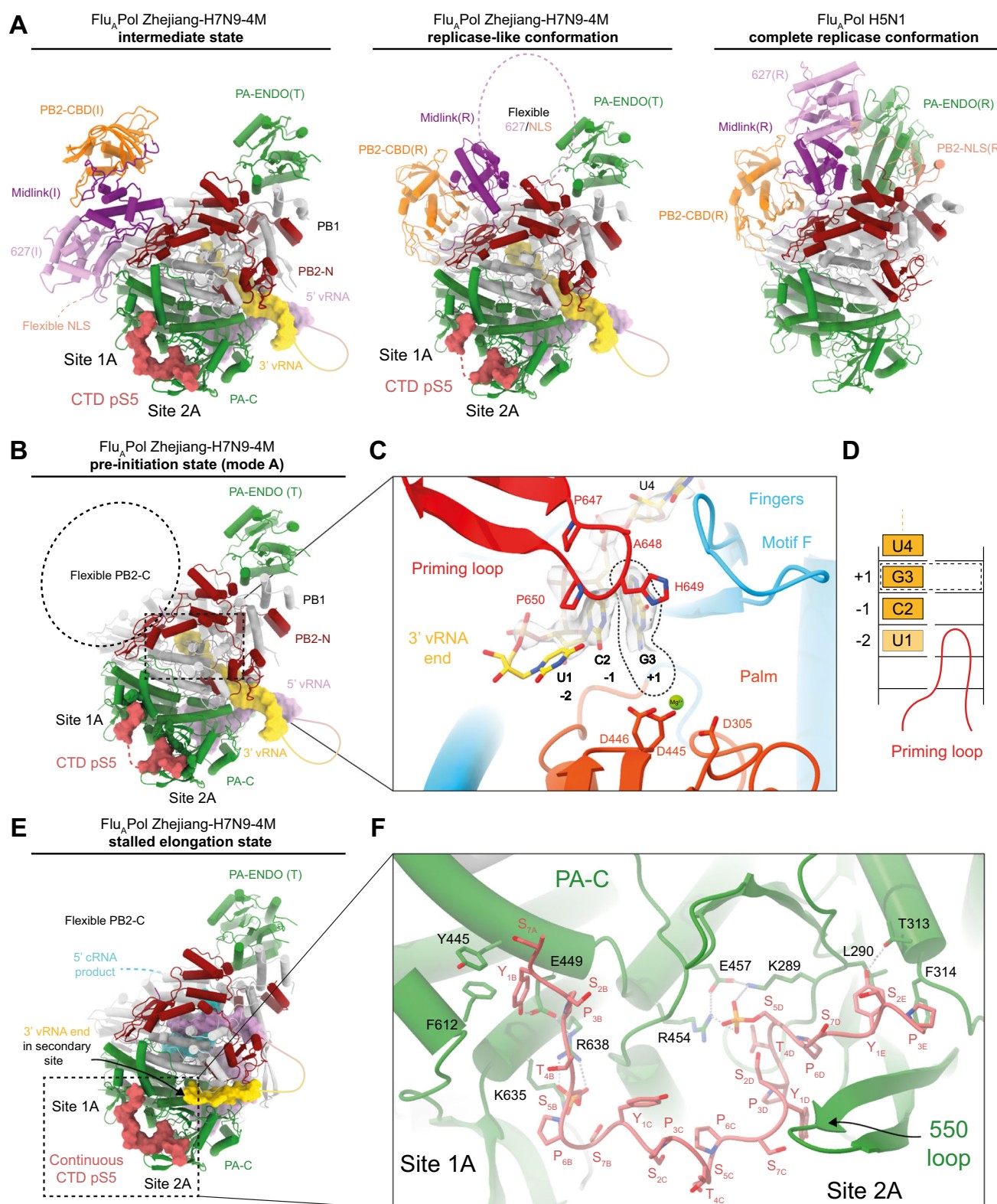

**A**

Flu$_A$Pol Zhejiang-H7N9-4M **intermediate state**

Flu$_A$Pol Zhejiang-H7N9-4M **replicase-like conformation**

Flu$_A$Pol H5N1 **complete replicase conformation**

**B** Flu$_A$Pol Zhejiang-H7N9-4M **pre-initiation state (mode A)**

**E** Flu$_A$Pol Zhejiang-H7N9-4M **stalled elongation state**

intermediate between the full replicase and transcriptase configurations, reflecting the flexibility of the peripheral PB2-C and PA-N domains (Fig. S7). Importantly, a replicase-like initiation complex (PB2-C(R)/ENDO(T)) is observed for the first time, characterised by the radically different position of the cap-binding domain compared to the transcriptase conformation (Fig. 4A, middle panel)[46], which was never observed during extensive studies of cap-dependent transcription[12]. However, the elongation state stalled with UpNHpp, in which the

translocated 3′ end of the template has reached the secondary binding site (Fig. 4E), is similar to the previously described pre-termination transcription state since PB2-C is not visible[12]. Finally, all structures have the CTD peptide mimic bound in both sites 1A and 2A, suggesting that both replication and transcription are consistent with CTD-binding (Fig. 4A, B and E). The observation of the intermediate state (denoted PB2-C(I)/ENDO(T)) in the context of de novo RNA synthesis lends support to the suggestion that it is an important intermediate in

**Fig. 4 | FluPol replication is consistent with CTD-binding to the FluPol. A** Left: Cartoon representation of Flu$_A$Pol Zhejiang-H7N9-4M in the intermediate conformation (obtained in this study) as previously described[14,45], bound to pS5 CTD. Middle: Flu$_A$Pol Zhejiang-H7N9-4M in a replicase-like conformation bound to pS5 CTD. Right: Flu$_A$Pol replicase conformation from A/duck/Fujian/01/2002(H5N1) (PDB: 6QPF)[41]. FluPols are aligned on the PB1 subunit. Flu$_A$Pol Zhejiang-H7N9-4M PA-ENDO remains in a transcriptase conformation (PA-ENDO(T)). In Flu$_A$Pol Fujian-H5N1 structure, it rotates and interacts with the PB2-NLS domain (PA-ENDO(R)). PA is coloured in green, PB1 in light grey, PB2-N in dark red, PB2-CBD in orange, PB2 mid-link in purple, PB2 627 in plum, PB2 NLS in salmon. The pS5 CTD is coloured in red, displayed as surface, with discontinuity between sites 1A/2A shown as a dotted line. PB2-627/NLS domains flexibility is highlighted as a dotted circle. RNAs are displayed as surfaces. The 5′ vRNA end is coloured in pink, the 3′ vRNA end in yellow. Flexible nts are represented as solid line. **B** Cartoon representation of Flu$_A$Pol Zhejiang-H7N9-4M structure in the pre-initiation state mode A. Colour code is identical to (**A**). PB2 C-terminal domains (PB2-C) are flexible, highlighted as a dotted circle. **C** Close-up view on the 3′ vRNA end in Flu$_A$Pol Zhejiang-H7N9-4M active site. The Coulomb potential map of the template is shown. 3′-U1 remains unseen in the map. The 3′-G3 is in the +1 active site, highlighted by a dotted line. The priming loop is coloured in red, the palm domain in orange with the catalytic aspartic acids displayed, coordinating a $Mg^{2+}$ ion, in green. The motif F is coloured in blue. **D** Schematic representation of the 3′ vRNA end terminal nucleotides active site position, as seen in (**C**). **E** Cartoon representation of Flu$_A$Pol Zhejiang-H7N9-4M in stalled elongation state. The colour code is identical to A. PB2-C domains are flexible. The cRNA de novo replication product is displayed as surface when visible or as a dotted line when flexible. The 3′ vRNA end is bound to the secondary site. **F** Close-up view on the pS5 CTD interaction with Flu$_A$Pol Zhejiang-H7N9-4M PA-C domain. FluPol residues interacting with the pS5 CTD are shown. Each CTD pS5 repeat is indicated.

the transition from transcriptase to replicase conformation[45]. Indeed, our structural results suggest that this transition might occur in the following sequence of steps: PB2-C(T)/ENDO(T), PB2-C(I)/ENDO(T), PB2-C(R)/ENDO(T), PB2-C(R)/ENDO(R), with the first and last being the full transcriptase and replicase, respectively. We suggest that the final step, ENDO(T) to ENDO(R) is more likely to happen after PB2-C(R) is established, which liberates the PB2-NLS domain so that it can interact with ENDO(R). Furthermore, we suggest that the full replicase conformation (including the otherwise flexible 627-domain) is only fully stabilised in the context of the replicase-encapsidase complex.

The pre-initiation state structures show that at least in vitro, and using highly purified recombinant FluPol, the preferred position of the 3′ end of the vRNA template is with C2, rather than U1, at the −1 position. This leads to formation of pppGpC at the beginning of the product, consistent with the NGS results, or alternatively, would allow efficient priming by pppApG, if this dinucleotide was already available from another source. Internal initiation at position 2 of vRNA has been previously described[20,47], but other authors report pppApG formation at position 1[7]. We note that initiation at position 2 of vRNA implies that the 5′ end of the cRNA product lacks A1 and thus probably forms a less stable hook structure since the A1:A10 non-canonical base pair cannot form. This could explain why we do not observe in cryo-EM the cRNA hook bound in a putative encapsidase, even though there is excess apo-polymerase. This could further explain why the putative asymmetric FluPol$_{(R)}$-FluPol$_{(E)}$ dimer is also not observed, since it seems likely that in vitro, for the Flu$_A$Pol, the FluPol$_{(R)}$-FluPol$_{(E)}$-ANP32 complex is not so stable, unlike the case of Flu$_C$Pol, where the FluPol$_{(R)}$-FluPol$_{(E)}$ complex is stable even in the absence of ANP32[13]. These observations suggest that other cellular or viral factors and the RNP context might be required to recapitulate true terminal initiation of vRNA replication and further work is required to clarify this issue[36].

### Restoration of CTD-binding interface rescues FluPol replication activity and enhances FluPol binding to huANP32A

Second-site mutations PA C489R and PB2 D253G were selected during passaging of the recombinant WSN virus with a PA K289A mutation (Fig. 2C) and conferred elevated FluPol-binding to the CTD as well as to huANP32A (Fig. 2F, grey and blue bars). PA C489 is in close proximity to the phosphoserine binding site in CTD-binding site 2A (Fig. 5A). The mutation PA C489R could therefore plausibly compensate for the loss of the positive charge of PA K289A and rescue the interaction with the CTD. Indeed, structural analysis by cryo-EM of Zhejiang-H7N9 WT FluPol in the symmetric dimeric form (Table 2 and Fig. S10) and bearing the double mutation PA K289A + C489R confirms that C489R points towards the phosphoserine binding site, although at a slightly greater distance than K289 (Fig. 5B, C). Despite this, the structure did not reveal CTD-binding to the Zhejiang-H7N9 PA K289A + C489R FluPol in this conformation. Therefore, we sought to analyse the

functional impact of the single PA C489R reversion on the viral phenotype, and in cellular assays, on FluPol activity and FluPol-binding to the CTD and huANP32A (Fig. 5D–I).

A recombinant WSN virus with the PA K289A + C489R mutations shows larger plaques compared to the PA K289A mutant virus, demonstrating that the PA C489R second-site mutation provides a significant growth advantage to PA K289A mutant virus (Fig. 5D and S11A). Indeed, WSN FluPol activity as measured in a vRNP reconstitution assay increases significantly when the PA C489R mutation is combined with PA K289A (Fig. 5E). This increase correlates with a restored mRNA:vRNA ratio (Fig. 5F and S11B) as well as an increased FluPol-binding to the CTD (Fig. 5G). Importantly, steady-state levels of the WT and mutant PA proteins are similar (Fig. S11C). These observations suggest that charge restoration in CTD-binding site 2A of the WSN FluPol in the transcriptase conformation rescues the interaction with the host RNAP II for 'cap-snatching' in cell-based assays and during live-virus infection. In cell-based assays, the Anhui-H7N9 FluPols with PA K289A and PA K289A + C489R show similar trends to those observed with the WSN FluPol counterparts, although with much less pronounced phenotypes (Fig. S11D). Therefore, we speculate that the fact that CTD-binding is not restored in vitro for Zhejiang-H7N9 PA K289A + C489R could be due to strain-specificities (Fig. S2) and/or to the presence of cellular factors that favour binding of the CTD to the PA K289A + C489R and are missing in the in vitro assay.

Beyond the rescue of FluPol transcription, cRNA accumulation levels increase significantly when PA C489R is combined with PA K289A (Fig. S11B), which goes together with a rescue of the imbalanced cRNA:vRNA ratio associated with the PA K289A mutant (Fig. 5H), and with increased huANP32A-binding (Fig. 5I). Taken together our observations suggest that the second-site mutation PA C489R has another functional impact, namely restoring FluPol replication by enhancing huANP32A-binding.

Similar results are obtained for the PA C453R revertant which has been shown to restore FluPol-binding to the CTD in site 1A by compensating the loss of positive charge of the PA R638A mutation (Fig. S12A)[23,48]. PA C453R rescues the attenuated viral plaque phenotype (Fig. S12B, C) as well as the reduced FluPol activity (Fig. S12D) associated with PA R638A, while it does not affect PA steady-state levels (Fig. S12E). Strikingly, PA C453R enhances huANP32A-binding when tested in combination with PA R638A (Fig. S12F), mimicking at CTD-binding site 1A the effects of PA C489R at site 2A.

The mechanism of rescue of ANP32A-binding by the PA C489R and PA C453R mutations remains unclear and could possibly involve the FluPol$_{(R)}$ and/or FluPol$_{(E)}$ moieties. The mutations could enhance binding of the pS5 CTD in site 1A or 2A in the FluPol$_{(R)}$, leading to an enhanced priming of the replicase activity in conjunction with ANP32. They could as well increase binding of ANP32 through direct interactions at the FluPol$_{(E)}$-ANP32 interface.

**Table 2 | Cryo-EM structures data collection, refinement and validation statistics (continued)**

| Name of structure | A/H7N9 K289A/C489R revertant Apo-dimer with promoter | A/H7N9 self-stalled termination product complex with bound CTD |
|---|---|---|
| PDB ID | PDB ID 8POH | PDB ID 8R3K |
| EMDB ID | EMD-17792 | EMD-18871 |
| **Data collection and processing** | | |
| Microscope | ThermoFisher Glacios | ThermoFisher Glacios |
| Voltage (kV) | 200 | 200 |
| Camera | Falcon IV/Selectris X | Falcon IV/Selectris X |
| Magnification | 130k | 130k |
| Nominal defocus range (µm) | −0.8/−2.0 | −0.8/−2.0 |
| Electron exposure (e−/Å²) | 40 | 40 |
| Number of fractions processed (no.) | 24 | 24 |
| Pixel size (Å) | 0.907 | 0.878 |
| Micrographs (no.) | 5092 | 3509 |
| **Refinement** | | |
| Particles per class (no.) | 278761 | 29072 |
| Map resolution (Å), 0.143 FSC | 3.26 | 3.43 |
| Model resolution (Å), 0.5 FSC | 3.38 | 3.40 |
| Map sharpening B factor (Å²) | −140 | −55 |
| Map versus model cross-correlation (CCmask) | 0.7456 | 0.8472 |
| **Model composition** | | |
| Non-hydrogen atoms | 20729 | 14287 |
| Protein residues | 2479 | 1718 |
| Nucleotide residues | 38 | 41 |
| Water | - | - |
| Ligands | 1 Mg | 2 Mg, PPi |
| **B factors (Å²)** | | |
| Protein | 105.84 | 55.88 |
| Nucleotide | 123.86 | 61.19 |
| Ligand | 33.48 | 44.02 |
| Water | - | - |
| **R.M.S. deviations** | | |
| Bond lengths (Å) | 0.003 | 0.003 |
| Bond angles (°) | 0.562 | 0.507 |
| **Validation** | | |
| MolProbity score | 1.84 | 1.44 |
| All-atom clash score | 6.79 | 6.44 |
| Poor rotamers (%) | 2.37 | 0.07 |
| **Ramachandran plot** | | |
| Favoured (%) | 96.91 | 97.59 |
| Allowed (%) | 3.09 | 2.41 |
| Outliers (%) | 0.0 | 0.0 |

## Discussion

### A model for RNAP II CTD-anchored transcription and replication of the influenza virus genome

There is long-standing and extensive evidence that binding of FluPol to the pS5 CTD repeats of host RNAP II is essential for transcription of viral mRNAs[26,49,50]. Here, we show that CTD-binding not only stabilises the transcriptase conformation of the influenza virus polymerase (FluPol$_{(T)}$)[22,29], but also enhances replication of the viral genome, in conjunction with the host protein ANP32A, which was recently described to bridge two FluPol moieties in an asymmetric replicase-encapsidase dimer (FluPol$_{(R)}$-FluPol$_{(E)}$)[13]. Our findings open new perspectives on the spatial coupling of viral transcription and replication and the coordinated balance between these two activities.

Early in infection, incoming vRNPs serve as a template for both primary mRNA transcription and the first round of cRNA synthesis[51]. Based on our findings, we propose a model in which incoming vRNPs switch from primary transcription to cRNA synthesis while remaining bound to the RNAP II CTD (Fig. 6A). It has been long understood that primary transcription requires FluPol$_{(T)}$ association with the RNAP II CTD to enable 'cap-snatching' for the priming of viral mRNA synthesis in the FluPol$_{(T)}$ conformation (Fig. 6B, C). Our structural and biochemical data reveal that a vRNA- and CTD-bound FluPol can adopt a FluPol$_{(R)}$ conformation and initiate vRNA to cRNA synthesis de novo, and that CTD-binding to the FluPol enhances cRNA synthesis. Importantly, during an ongoing infection the switch from a transcribing FluPol$_{(T)}$ to a replicating FluPol$_{(R)}$ can occur only after de novo synthesised FluPol and NP are imported into the nucleus (Fig. 6D) and an asymmetric FluPol$_{(R)}$-ANP32A-FluPol$_{(E)}$ complex is assembled, ensuring encapsidation of the nascent viral RNA by the FluPol$_{(E)}$ moiety in conjunction with the NP. Our cell-based and in vitro assays show that the RNAP II CTD and ANP32 jointly enhance cRNA synthesis. The data support a model in which the replicating FluPol$_{(R)}$ remains associated to pS5 CTD repeats while being assembled into a FluPol$_{(R)}$-ANP32A-FluPol$_{(E)}$ complex (Fig. 6E, F). The CTD is a low complexity disordered region and there is evidence that it drives phase separation and the formation of biomolecular condensates that concentrate RNAP II and transcription regulatory factors[52]. In infected cells, the RNPA II condensates may in addition concentrate the FluPol and its viral and cellular partners, allowing CTD-bound FluPols to switch efficiently between transcription or replication depending on the relative availability of, e.g. 5' capped RNAs, NP and/or apo FluPol (Fig. 6G), and perform both activities in a coordinated manner.

Although a structure for the asymmetric FluPol$_{(R)}$-ANP32-FluPol$_{(E)}$ replicase-encapsidase dimer has so far been obtained only for vRNA-bound Flu$_C$Pol$_{(R)}$, it is generally thought that the same type of asymmetric dimer is formed by a cRNA-bound FluPol$_{(R)}$[18]. Therefore, the model proposed for incoming vRNPs (Fig. 6) may possibly be extended to neo-synthesised progeny vRNPs or cRNPs, consistent with vRNA synthesis being detected mostly in the chromatin-associated fractions[53,54]. In our model, RNPs could, therefore, remain anchored to the same CTD repeats and undergo several rounds of transcription (vRNPs) or genome replication (vRNPs or cRNPs). Progeny RNPs might in turn associate to the same or a close CTD, thereby generating a viral factory for RNA synthesis. Further studies will be needed to investigate the precise relationships between replicating vRNP, cRNPs and host RNAP II, given that cRNPs do not perform 'cap-snatching' and require transactivation by an apo FluPol, and could therefore be subjected to different regulatory networks[21,41].

Our model that the RNAP II CTD provides a recruitment platform for the different types of monomeric and multimeric polymerases raises the question of how the CTD is coordinating transcription versus replication in the context of chromatin in infected cells. Influenza infection was shown to strongly dysregulate RNPA II activities, as evidenced by a decreased RNAP II occupancy in gene bodies downstream of the transcription start site[49,55], a widespread RNAP II termination defect and subsequent transcriptional read-through[55–57], as well as a proteolytic degradation of the RNAP II at late stages of infection[58,59]. FluPol-binding to the CTD is dependent on S5 phosphorylation[22,23,26], and our data show that pS5 CTD-binding is compatible with both transcription and replication activity of the FluPol. Consistently, a large proportion of vRNPs were found to colocalize with pS5 RNAP II at

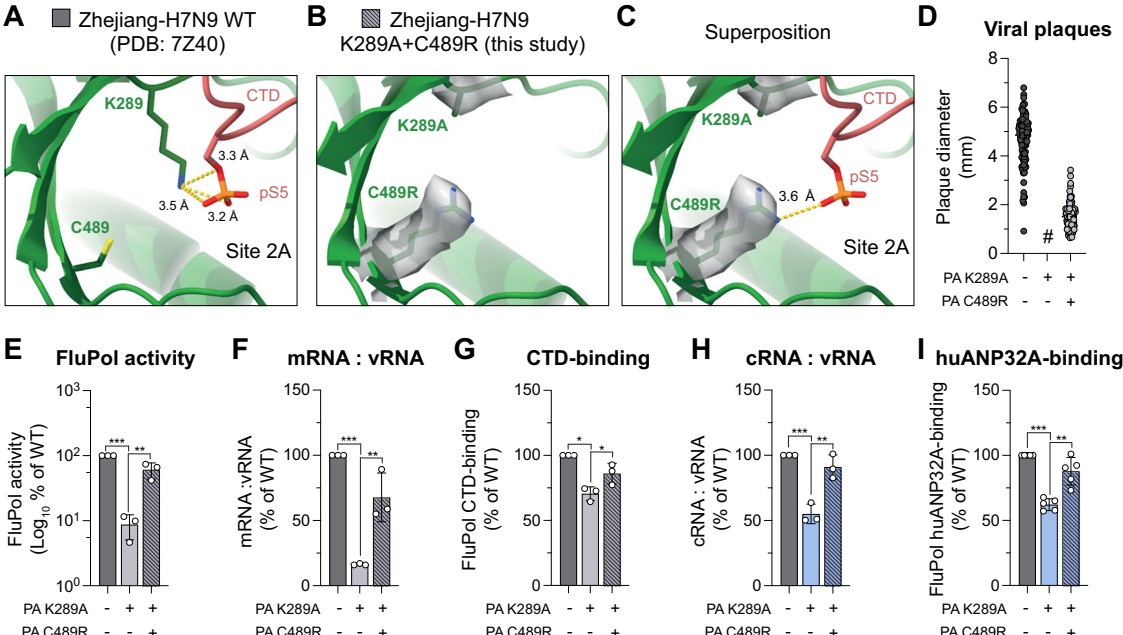

**Fig. 5 | Restoration of CTD-binding interface rescues FluPol replication activity and enhances FluPol binding to huANP32A. A** Cartoon representation of the pS5 CTD-bound to FluPol Zhejiang-H7N9 in site 2A (PDB: 7Z4O)[22]. PA subunit is coloured in green, pS5 CTD in red. PA K289-C489 residues are displayed. Putative hydrogen bonds are drawn as yellow dashed lines. Distances are indicated. **B** Cartoon representation of Flu$_A$Pol Zhejiang-H7N9 PA K289A + C489R (obtained in this study). The Coulomb potential map of PA/K289A-C489R is shown. **C** Superposition of the pS5 CTD, extracted from the structure shown in (**A**), with the Flu$_A$Pol Zhejiang-H7N9 PA K289A + C489R structure shown in (**B**). The putative hydrogen bond between PA C489R and pS5 is shown. Distance is indicated. **D**–**I** Phenotypes associated with the WSN FluPol PA K289A primary mutation and PA C489R second-site mutation. **D** Plaque phenotype of recombinant WSN mutant viruses produced by reverse genetics ($n = 2$), analyzed as in Fig. 2E (see Fig. S11A) (#) pinhead-sized plaques. **E** WSN FluPol activity was measured by vRNP reconstitution

in HEK-293T cells, using a model Fluc-vRNA. Luminescence signals are represented as a percentage of PA WT. **F** WSN vRNPs were reconstituted in HEK-293T cells using the NA vRNA segment. The steady-state levels of NA mRNA and vRNA were quantified by strand-specific RT-qPCR[35], normalised to GAPDH by the $2^{-\Delta\Delta CT}$ method[71] and are presented as ratios of mRNA to vRNA levels relative to PA WT (RNA levels are shown in Fig. S11B). **G** WSN FluPol binding to the CTD was assessed using a split-luciferase-based complementation assay. Luminescence signals are represented as a percentage of PA WT. **H** Accumulation levels of cRNA and vRNA in a vRNP reconstitution assay were determined by strand-specific RT-qPCR[35] as in (**F**). Ratios of cRNA to vRNA levels relative to PA WT are shown (RNA levels are shown in Fig. S11B). **I** huANP32A-binding to WSN FluPol was determined as in (**G**). (mean ± SD, $n = 3, 3, 3, 5$, *$p < 0.033$, **$p < 0.002$, ***$p < 0.001$, one-way ANOVA; Dunnett's multiple comparisons test). Source data are provided as a Source data file.

6 hours post-infection, a stage when viral replication is becoming predominant over transcription[55]. S5 phosphorylation is considered as the hallmark modification of RNAP II initiation[28]. Because there is a short time window for the influenza polymerase to gain access to capped nascent RNAP II transcripts before the high affinity nuclear cap-binding complex binds the capped RNA, 'cap-snatching' is thought to occur in association with promoter-proximal paused RNAP II[22–24]. However, recent studies show that pS5 CTD can also occur during RNAP II elongation[60], and it remains unknown to what extent the CTD phosphorylation dynamics is altered upon influenza infection. Therefore, in principle, pS5 CTD-bound FluPol could perform viral replication in association with initiating and/or elongating RNAP II. Investigating whether or not transcribing and replicating FluPol complexes are associated to the same phosphoforms of the CTD, in terms of complex phosphorylation patterns of residues $S_2$, $S_5$, $S_7$ but also $Y_1$ and $P_4$[61], and/or localised to the same RNAP II condensates, could help address this question. Moreover, further investigation are required to gain a better understanding of how FluPol transcription and replication complexes interact with the full-length CTD and other RNAP II domains, and to what extent the CTD and/or host factors recruited by the CTD are controlling the balance between transcription and replication. Whether the viral-induced degradation of RNAP II observed at late time-points during influenza infection[58,59] could favour the release of neo-synthesised vRNPs from the chromatin and their nuclear export also warrants investigation. Fully addressing these questions will require to overcome technical challenges such as structural analyses on the FluPol in the context of vRNPs and cRNPs, or high-resolution

imaging in live-infected cells to visualise the subnuclear localisation of FluPol transcription and replication.

## Methods

### Cells
HEK-293T cells (ATCC CRL-3216 and ATCC CRL-11268) were grown in complete Dulbecco's modified Eagle's medium (DMEM, Gibco) supplemented with 10% foetal bovine serum (FBS) and 1% penicillin-streptomycin (Gibco). MDCK cells (provided by the National Influenza Center, Paris, France) were grown in Modified Eagle's medium (MEM, Gibco) supplemented with 5% FBS and 1% penicillin-streptomycin.

### Plasmids
A/WSN/33 (WSN) ORFs were encoded in pcDNA3.1-PB2, -PB1, -PA and pCI-NP plasmids[22,23]. The A/Anhui/1/2013 (Anhui-H7N9) pCAGGS-PB2, -PB1, -PA and NP plasmids were kindly provided by R. Fouchier (Erasmus Medical Center, Netherlands)[62]. The B/Memphis/13/2003 ORFs were encoded in pcDNA3.1-PB2 and PA plasmids[9]. The WSN reverse genetics plasmids were kindly provided by G. Brownlee (Sir William Dunn School of Pathology, Oxford, UK)[63]. For vRNP reconstitution assays, a pPolI-Firefly plasmid encoding the Firefly luciferase sequence in negative polarity flanked by the 5' and 3' non-coding regions of the IAV NS segment was used and the pTK-Renilla plasmid (Promega) was used as an internal control. For RNA quantifications in vRNP reconstitution assays by strand-specific qRT-PCR a pPR7-WSN-NA reverse genetic plasmid was used[64]. For the WSN and Anhui-H7N9 pCI-PB1-G1 and pCI-PB1-G2 plasmids used for G. princeps split-luciferase-based

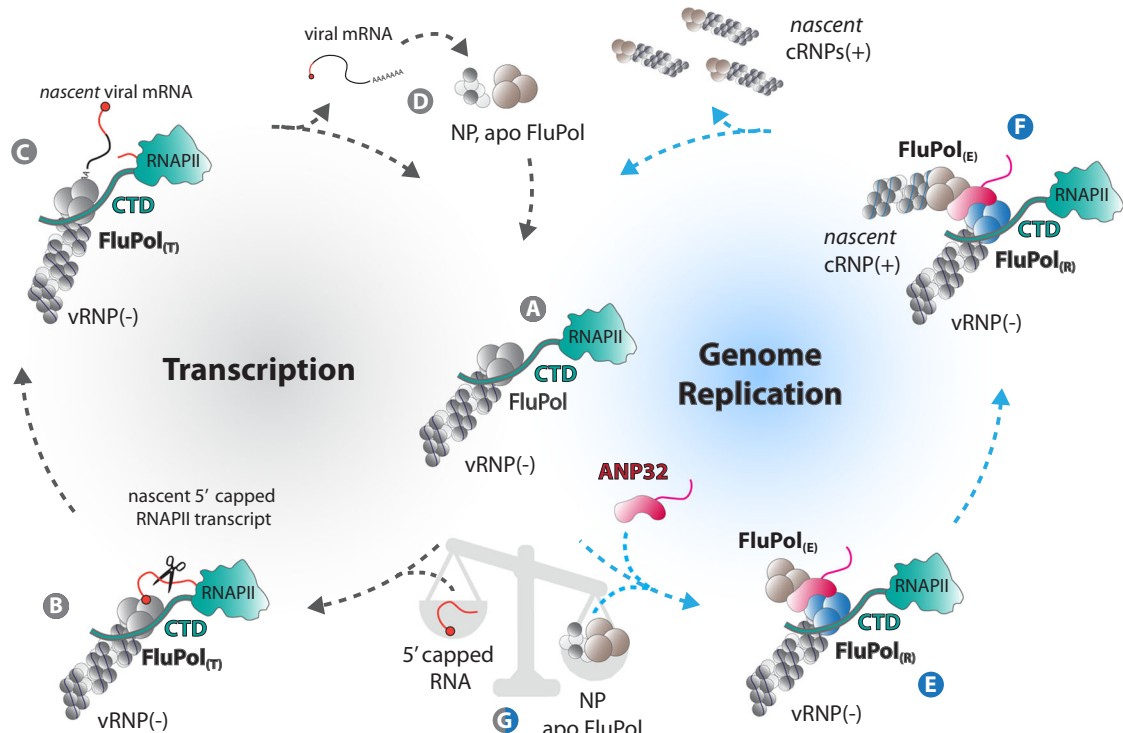

**Fig. 6 | A model for RNAP II CTD-anchored transcription and replication of the influenza virus genome.** A model for RNAP II CTD-anchored transcription and replication of the influenza virus genome. Upon influenza virus infection, incoming vRNPs are imported into the nucleus and bind to the host RNAP II CTD through bipartite interaction sites on the FluPol (**A**). This intimate association allows FluPol, in the transcriptase conformation (FluPol(T)), to cleave short capped oligomers derived from nascent RNAP II transcripts in a process referred to as 'cap-snatching' to initiate primary transcription of viral mRNAs (**B**). Polyadenylation is achieved by a non-canonical mechanism involving stuttering of the viral polymerase at a poly-adenylation signal (**C**). The 5′ and 3′ vRNA extremities always remain bound to the polymerase which allows efficient recycling from the termination to the initiation state (**C**–**A**). Upon translation of viral mRNAs, de novo synthesised FluPols in an apo state (not viral RNA-bound) and NPs are imported into the nucleus (**D**). The apo

FluPol, in conjunction with the host factor ANP32, associates with the parental CTD-associated FluPol(T) and triggers its conformational transition into a replicating FluPol(R), to form an asymmetric FluPol(R)-FluPol(E) dimer (**E**) where FluPol(E) is encapsidating the newly synthesised cRNA in conjunction with NP (**F**). The FluPol(R)-ANP32-FluPol(E) replication complex remains associated to the RNAP II through direct binding of the CTD to FluPol(R). Anchoring of the parental vRNP to the CTD allows it to engage into successive cycles of either viral genome replication or mRNA transcription, depending on the availability of NP, apo FluPol and/or nascent capped oligomers derived from actively transcribing RNAP II (**G**). Such switching between both activities allows efficient adaptation to waving levels of de novo synthesised vRNP components in the nucleus of an infected cell and is key to ensure a correct balance between genome replication and mRNA transcription.

protein-protein complementation assays, the PB1 ORF was tagged C-terminally with the *G. princeps* fragments[65]. The Anhui-H7N9 PA open-reading frame (ORF) was subcloned into the pCI vector and the PA-X ORF was deleted when indicated by introducing silent mutations at the site of ribosomal frameshifting[66]. For CTD-binding assays, a C-terminal stretch of RPB1 (108 amino acids) in conjunction with 14 CTD repeats (termed CTD) were tagged C-terminally with the *G. princeps* fragments[34]. For ANP32A-binding assays, ANP32A was tagged C-terminally with the *G. princeps* fragments[67]. The pCI-mCherry-CTD-WT and mCherry-CTD-S5A plasmids were generated by replacing the G2 sequence in pCI-G2-CTD constructs[22]. ANP32A encoding the amino acids 1-249 (full-length), 1-199 and 1-149 were subcloned into the pCI vector with a C-terminal V5-tag and SV40 nuclear localisation signals (NLS). DDX5 and RED ORFs were encoded in the pCI vector and tagged N-terminally with the *G. princeps* fragments[38,68]. All mutations were introduced by an adapted QuikChange site-directed mutagenesis (Agilent Technologies) protocol[69]. The ORFs were verified by Sanger sequencing. Plasmid sequences are provided as a source data file and are available with annotations at [https://doi.org/10.5281/zenodo.10462746].

### Generation of HEK-293T ANP32A and ANP32B knockout cells

CRISPR-Cas9-mediated HEK-293T (ATCC CRL-11268) ANP32A and ANP32B knockout cells (HEK-293T ANP32AB KO) and a control cell line

(HEK-293T CTRL) were generated by lentiviral transduction. Guide RNAs (gRNAs) targeting ANP32A (5′-ACCGTCAGGTGAAAGAACTTGTCC-3′ and 5′-AAACGGACAAGTTCTTTCACCTGA-3′), ANP32B (5′-ACCGGGAGCTGAGGAACCGGACCC-3′ and 5′-AAACGGGTCCGGTTCCTCAGCTCC-3′) and a non-targeting control (5′-ACCGGTATTACTGATATTGGTGGG-3′ and 5′-AAACCCCACCAATATCAGTAATAC-3′) were annealed and cloned into the BsmBI site of pSicoR-CRISPR-PuroR (a kind gift from Robert Jan Lebbink[70]). Replication-incompetent lentiviral particles were generated by transient transfection of HEK-293T cells with the pSicoR-CRISPR-PuroR plasmid harbouring gRNAs targeting either ANP32A, ANP32B or a non-targeting control gRNA as well as with the packaging plasmids pMD2.G and psPAX2 (gifts from Didier Trono, Addgene plasmids #12259 and #12260, respectively). Supernatants were harvested 72 hpt, centrifuged to remove cellular debris and passed through 0.45 μm sterile filters. Fresh HEK-293T cells were plated in 6 well plates precoated with poly-D-lysine and were infected with 3 ml of lentiviral supernatant by constant centrifugation at 700 g for 40 min in the presence of 10 μg/ml Polybrene (Sigma-Aldrich). At 72 hpi, the medium was exchanged to selection medium containing 1 μg/ml puromycin (PAA Laboratories) for one week. Surviving cells were then cloned by limiting dilution in 96-well plates. Individual cell clones were grown and ANP32A and ANP32B protein levels in cellular lysates of individual clones was assessed by immunoblotting. To generate HEK-293T cells with a double KO for ANP32A and ANP32B, two rounds of targeting were performed.

## Protein complementation assays

HEK-293T cells were seeded in 96-well white plates (Greiner Bio-One) the day before transfection using polyethyleneimine (PEI-max, #24765-1 Polysciences Inc). Cells were lysed 20-24 hpt in Renilla lysis buffer (Promega) for 45 min at room temperature (RT) under steady shaking. G. princeps luciferase enzymatic activity due to luciferase reconstitution was measured on a Centro XS LB960 microplate luminometer (Berthold Technologies, MikroWin Version 4.41) using a reading time of 10 s after injection of 50 µl Renilla luciferase reagent (Promega). Mean relative light units (RLUs) of technical triplicates of G. princeps split-luciferase-based protein-protein complementation assays are represented. In the graphs each dot represents an independently performed biological replicate while at least three experiments were performed in each case. For FluPol-binding assays cells were always co-transfected with plasmids encoding the viral polymerase subunits (PB2, PB1, PA) tagged to one part of the G. princeps luciferase and the target protein tagged to the other part. For FluPol oligomerisation PB2, PA, PB1-G1 and PB1-G2 were co-transfected and when indicated an expression plasmid for mCherry, mCherry-CTD-WT or mCherry-CTD-S5A was added while the total amount of transfected plasmid was adjusted using an empty control plasmid. Plasmid combinations, orientations of tags as well as plasmid quantities used for transfections in a given experiment are available as a source data file.

## vRNP reconstitution assays

HEK-293T cells were seeded in 96-well white plates (Greiner Bio-One) the day before transfection. Cells were co-transfected with plasmids encoding the vRNP protein components (PB2, PB1, PA, NP), a pPolI-Firefly plasmid encoding a negative-sense viral-like RNA expressing the Firefly luciferase and the pTK-Renilla plasmid (Promega) as an internal control. For FluPol trans-complementation assays, equal amounts of expression plasmids expressing the trans-complementing PA mutants were co-transfected. The fold-change shown for FluPol trans-complementation assays represents the increase above the integrated background of the respective FluPol mutants alone. For FluPol activity rescue experiments in HEK-293T ANP32AB KO cells, expression plasmids encoding FL or truncated versions of ANP32A were co-transfected. Firefly luciferase activity due to viral polymerase activity and Renilla luciferase activity due to RNAP II activity were measured using the Dual-Glo Luciferase Assay system (Promega) according to the manufacturer's instructions. The graphs represent viral polymerase activity normalised to RNAP II activity. Luciferase activities were measured in technical duplicates at 20–24 hpt or at 48 hpt when indicated. In the graphs each dot represents an independently performed biological replicate while at least three experiments were performed in each case. For the quantification of mRNA, cRNA and vRNA levels, HEK-293T cells were seeded in 12-well plates and transfected with plasmids encoding the vRNP protein components (PB2, PB1, PA, NP) and a low amount of WSN-NA RNA expressing plasmid (5 ng/well). Total RNA was isolated at 24 hpt with RNeasy Mini columns according to the manufacturer's instructions (RNeasy Kits, Qiagen) and strand-specific RT-qPCRs were performed[35]. Briefly, total RNA was reverse transcribed using primers specific for NA mRNA, cRNA, vRNA and, when indicated, the cellular glyceraldehyde 3-phosphate dehydrogenase (GADPH) with SuperScript™ III Reverse Transcriptase (Invitrogen) and quantified using SYBR-Green (Roche) with the Light-Cycler 480 system® (Roche, Light Cycler Software 1.5.0.39). RNA levels were normalised to GAPDH when indicated and analysed using the $2^{-\Delta\Delta CT}$ as described before[71]. Plasmid combinations and plasmid quantities used for transfections in a given experiment are available as a source data file.

## Antibodies and immunoblots

Total cell lysates were prepared in RIPA cell lysis buffer[72]. Proteins were separated by SDS-PAGE using NuPAGE™ 4-12% Bis-Tris gels (Invitrogen) and transferred to nitrocellulose membranes which were incubated with primary antibodies directed against pS5 CTD (clone 3E8, Active Motif, 1:1000), mCherry (26765-1-AP, Proteintech, 1:000), V5 (SV5-Pk1, Thermo Fisher, 1:5000), Tubulin (B-5-1-2, Sigma-Aldrich, 1:10000), PA[73] (1:2500), PB2 (GTX125925, GeneTex, 1:5000), ANP32A (AV40203, Sigma-Aldrich, 1:2500), ANP32B (EPR14588, AbCam, 1:2500) and subsequently with HRP-tagged secondary antibodies (Jackson Immunoresearch). Membranes were revealed with the ECL2 substrate according to the manufacturer's instructions (Pierce) and chemiluminescence signals were acquired using the ChemiDoc imaging system (Bio-Rad) and analysed with ImageLab (Bio-Rad). Uncropped gels are provided as a source data file.

## Production and characterisation of recombinant viruses

The recombinant PA mutant influenza viruses (PA K289A, R454A, K635A, R638A) used for serial cell culture passaging were generated by reverse genetics followed by plaque purification under agarose overlay and one round of viral amplification (P1)[23]. For the serial passaging, three dilutions of the supernatant collected at passage N (1:10, 1:100 and 1:1000) were used to infect MDCK cells at passage N + 1. The supernatant corresponding to one of these dilutions (the lowest dilution that resulted in the infectious titre greater than or equal to titre at passage N) was selected to infect MDCK cells at passage N + 2. These conditions resulted in MOIs in the 0.001 or 0.0001 range, depending on the passages. Upon infection, MDCK cells were incubated for 3 days at 37 °C in DMEM containing TPCK-Trypsin (Sigma) at a final concentration of 1 µg/mL. Viral supernatants were titrated on MDCK cells in a plaque assay[74]. Viral RNA was extracted from 140 µL of viral stocks using the QIAamp Viral RNA Mini kit (Qiagen) according to the manufacturer's instructions. For next-generation sequencing of the full viral genome reverse transcription and amplification of the eight genomic segments were performed using the RT-qPCR protocol adapted by the National Influenza Center (Institut Pasteur)[75]. Next-generation sequencing was performed by the P2M facility at Institut Pasteur using the Nextera XT DNA Library Preparation kit (Illumina), the NextSeq 500 sequencing systems (Illumina) and the CLC Genomics Workbench 9 software (Qiagen) for analysis. Recombinant viruses with a selected subset of observed second-site mutations were generated by reverse genetics with an adapted protocol[76]. In brief, HEK-293T cells were seeded in 6-well plates the day before transfection with the WSN reverse genetics plasmids[63] in Opti-MEM (Gibco) using FuGene®6 (Promega) according to the manufacturer's instructions. The next day, MDCK were added to the wells in 500 µL Opti-MEM containing 0.5 µg/mL TPCK trypsin (Sigma). The following days, 500 µL Opti-MEM containing 1 and 2 µg/mL TPCK trypsin, respectively, were added. One day later, the supernatants were harvested, centrifuged and stored at −80 °C. The reverse genetics supernatants were titrated on MDCK cells and plaque diameters were measured upon staining with crystal violet using Fiji (ImageJ2 Version 2.3.0/1.3q)[77].

## Influenza virus polymerase Zhejiang-H7N9 (WT, 4 M, PA K289A + C489R)

The pFastBac Dual vector encoding for the A/Zhejiang/DTID-ZJU01/2013 (H7N9) polymerase heterotrimer subunits, PA (Uniprot: M9TI86), PB1 (Uniprot: M9TLW3), and PB2 (Uniprot: X5F427) was used as a starting point (Zhejiang-H7N9 WT)[43]. The mutations PA E349K, PA R490I, PB1 K577G and PB2 G74R were introduced by a combination of PCRs and Gibson assembly (Zhejiang-H7N9 4 M). A similar approach has been applied to clone the FluPol Zhejiang-H7N9 PA K289A + C489R mutant. Plasmid sequences were confirmed by sanger sequencing for each polymerase subunit. All primer and plasmid sequences are available upon request.

FluPols Zhejiang-H7N9 (WT/4M/PA K289A + C489R) were produced using the baculovirus expression system in Trichoplusia ni High five cells. For large-scale expression, cells at 0.8-1E06 cells/mL

concentration were infected by adding 1% of virus. Expression was stopped 72 to 96 h after the day of proliferation arrest and cells were harvested by centrifugation (1.000 g, 20 min at 4 °C).

Cells were disrupted by sonication for 4 min (5 s ON, 20 s OFF, 40% amplitude) on ice in lysis buffer (50 mM HEPES pH 8, 500 mM NaCl, 0.5 mM TCEP, 10% glycerol) with cOmplete EDTA-free Protease Inhibitor Cocktail (Roche). After lysate centrifugation at 48.384 g during 45 min at 4 °C, ammonium sulphate was added to the supernatant at 0.5 g/mL final concentration. The recombinant protein was then collected by centrifugation (45 min, 4 °C at 70.000 g), resuspended in the lysis buffer, and the procedure was repeated another time. FluPol Zhejiang-H7N9 was then purified using His60 NiNTA Superflow resin (Takara Bio) from the soluble fraction. Bound proteins were subjected to two sequential washes using (i) the lysis buffer supplemented by 1 M NaCl and (ii) the lysis buffer supplemented by 50 mM imidazole. Remaining bound proteins were eluted using the lysis buffer supplemented by 500 mM imidazole. Fractions with FluPol Zhejiang-H7N9 were pooled and directly subjected to a strep-tactin affinity purification (IBA, Superflow). Bound proteins were eluted using the lysis buffer supplemented by 2.5 mM d-desthiobiotin and protein-containing fractions were pooled and diluted with an equal volume of buffer (50 mM HEPES pH 8, 0.5 mM TCEP, 10% glycerol) before loading on a third affinity column HiTrap Heparin HP 5 mL (Cytiva). A continuous gradient of lysis buffer supplemented with 1 M NaCl was applied over 15 CV, and FluPol Zhejiang-H7N9 was eluted as single species at ~800 mM NaCl. Pure and nucleic acid free FluPol Zhejiang-H7N9 was dialysed overnight in a final buffer (50 mM HEPES pH 8, 500 mM NaCl, 2 mM TCEP, 5% glycerol), concentrated with Amicon Ultra-15 (50 kDa cutoff), flash-frozen and stored at −80 °C for further use.

### Influenza virus nucleoprotein Anhui-H7N9 monomeric mutant (R416A)

The influenza virus NP Anhui-H7N9 gene was amplified from a pCAGGS plasmid (gift from Ron Fouchier)[62] and introduced by Gibson assembly in a pLIB vector with an N-terminal double strep-tag followed by a human rhinovirus (HRV) 3C protease cleavage site. The NP R416A mutation was introduced by a combination of PCRs and Gibson assembly. Introduction of the desired mutation was confirmed by sanger sequencing. Monomeric influenza virus Anhui-H7N9 NP R416A was produced using the baculovirus expression system in Trichoplusia ni High five cells. For large-scale expression, cells at 0.8-1E06 cells/mL concentration were infected by adding 1% of virus. Expression was stopped 72–96 h after the day of proliferation arrest and cells were harvested by centrifugation (1.000 g, 20 min at 4 °C).

Cells were disrupted by sonication for 4 min (5 s ON, 15 s OFF, 40% amplitude) on ice in lysis buffer (50 mM HEPES pH 8, 500 mM NaCl, 2 mM TCEP, 5% glycerol) with cOmplete EDTA-free Protease Inhibitor Cocktail (Roche). After lysate centrifugation at 48.384 g for 45 min at 4 °C, the soluble fraction was loaded on a StrepTrap HP 5 mL (Cytiva). Bound proteins were eluted using the lysis buffer supplemented by 2.5 mM d-desthiobiotin. Protein-containing fractions were pooled and diluted with an equal volume of buffer (50 mM HEPES pH 8, 2 mM TCEP, 5% glycerol) before loading on an affinity column HiTrap Heparin HP 5 mL (Cytiva). A continuous gradient of lysis buffer supplemented with 1 M NaCl was applied over 15 CV, and Anhui-H7N9 NP R416A was eluted as single species at ~500 mM NaCl without nucleic acids ($A_{260/280}$: 0.6). Pure and nucleic acid free Anhui-H7N9 NP R416A was dialysed overnight in a final buffer (50 mM HEPES pH 8, 300 mM NaCl, 1 mM TCEP, 5% glycerol) together with N-terminal his-tagged HRV 3C protease (ratio 1:5 w/w). Tag-cleaved Anhui-H7N9 NP R416A was subjected to a last Ni-sepharose affinity to remove the HRV 3C protease, further concentrated with Amicon Ultra-15 (10 kDa cutoff), flash-frozen and stored at −80 °C for later use.

### Acidic nuclear phosphoprotein 32A

Human and chicken ANP32A genes (GeneScript) were introduced in a pETM11 vector with an N-terminal 6xHis-tag followed by a Tobacco Etch Virus (TEV) protease cleavage site. ANP32A constructs were expressed in BL21(DE3) E.coli cells. Expression was induced when absorbance reached 0.6, with 1 mM IPTG, incubated for 4 h at 37 °C. Cells were harvested by centrifugation (1.000 g, 20 min at 4 °C).

Cells were disrupted by sonication for 5 min (5 s ON, 15 s OFF, 50% amplitude) on ice in lysis buffer (50 mM HEPES pH 8, 150 mM NaCl, 5 mM beta-mercaptoethanol (BME)) with cOmplete EDTA-free Protease Inhibitor Cocktail (Roche). After lysate centrifugation at 48.384 g for 45 min at 4 °C, the soluble fraction was loaded on a HisTrap HP 5 mL column (Cytiva). Bound proteins were subjected to a wash step using the lysis buffer supplemented by 50 mM imidazole. Remaining bound proteins were eluted using the lysis buffer supplemented by 500 mM imidazole. Fractions containing ANP32A were dialysed overnight in the lysis buffer (50 mM HEPES pH 8, 150 mM NaCl, 5 mM BME) together with N-terminal his-tagged TEV protease (ratio 1:5 w/w). Tag-cleaved ANP32A protein was subjected to a Ni-sepharose affinity column to remove the TEV protease, further concentrated with Amicon Ultra-15 (3 kDa cutoff) and subjected to a Size-Exclusion Chromatography using a Superdex 200 Increase 10/300 GL column (Cytiva) in a final buffer containing 50 mM HEPES pH 8, 150 mM NaCl, 2 mM TCEP. Fractions containing exclusively ANP32A were concentrated with Amicon Ultra-15 (3 kDa cutoff), flash-frozen and stored at −80 °C for later use.

Human ANP32A truncation constructs (1–199 and 144–249) were generated, expressed and protein purified as previously described ref. 78. The human ANP32A 1-149 construct was a gift from Cynthia Wolberger (Addgene plasmid #67241[79]) and was expressed and protein purified as previously described[78].

### De novo FluPol replication activity

Synthetic vRNA loop ('v51_mut_S') (5'-pAGU AGA AAC AAG GGU GUA UUU UCC CCU CUU UUU GUU UCC CCU GCU UUU GCU -3') (IDT) was used for all in vitro replication activity assays. For all de novo replication activity assays, 2.4 µM FluPol Zhejiang-H7N9 (WT or 4 M) were mixed with (i) 0.8 µM v51_mut_S and/or (ii) 8 µM ANP32A and/or (iii) 16 µM pS5 CTD(6mer) (respective molar ratio: 3 FluPols: 1 template: 10 ANP32: 20 pS5 CTD). Reactions were launched at 30 °C for 4 h by adding ATP/GTP/CTP/UTP (AGCU) or only ATP/GTP/CTP (AGC), α-32P ATP (PerkinElmer) and MgCl₂ in a final assay buffer containing 50 mM HEPES pH 8, 150 mM NaCl, 2 mM TCEP, 100 µM/NTP, 1 mM MgCl₂, 0.05 µCi/µl α-32P ATP. Reactions were stopped by adding 2X RNA loading dye, heating 5 min at 95 °C and immediately loaded on a 20% TBE-7M urea-polyacrylamide gel. Each gel was exposed on a storage phosphor screen and read with an Amersham Typhoon scanner (Cytiva). For each gel the decade markers system (Ambion) was used.

### Next generation sequencing of in vitro FluPol replication products

To confirm the identity of the replication products (full-length cRNA product and stalled cRNA product) and address the discrepancy between expected size and urea-PAGE migration, sequencing of the total reaction product was performed from a de novo FluPol replication assay. 2.4 µM FluPol Zhejiang-H7N9 4 M were mixed with (i) 0.8 µM v51_mut_S, (ii) 8 µM ANP32A and (iii) 16 µM pS5 CTD(6mer) (respective molar ratio: 3 FluPols: 1 template: 10 ANP32A: 20 pS5 CTD). Reaction was launched at 30 °C for 4 h by adding ATP/GTP/CTP/UTP and MgCl₂ in a final assay buffer containing 50 mM HEPES pH 8, 150 mM NaCl, 2 mM TCEP, 100 µM/NTP, 1 mM MgCl₂. After completion of the reaction, proteins were removed using the Monarch RNA Cleanup Kit (NEB) and 5'-mono-phosphorylated v51_mut_S templates were specifically digested with a terminator 5'-phosphate-dependent exonuclease (Biosearch technologies) and subjected again to the

Monarch RNA Cleanup Kit (NEB). Total remaining RNAs were used for next-generation sequencing.

Sample concentration and fragment size distribution were checked with the RNA Pico assay (Bioanalyzer, Agilent). 6 ng of sample was treated with 2 units T4 Polynucleotide Kinase (NEB) and incubated at 37 °C for 30 min, followed by heat inactivation at 65 °C for 20 min. The treated samples were then taken into library preparation following the Takara SMARTer small RNA sequencing kit according to the manufacturer's instructions with 13 cycles of PCR. The final library fragment size distribution was checked with the High Sensitivity Bioanalyzer assay. 8 pM of library was sequenced on a MiSeq to generate 76 bp single-end reads.

Resulting total reads (13.073.371) were trimmed using Cutadapt v2.3[80] with settings recommended by the library preparation kit (-u 3 -a 'AAAAAAAAAA') omitting the filter for only reads longer than 20 bases (10.991.864). Trimmed reads were subsequently used as an input for pattern lookup. All reads exactly starting by these motifs have been kept for further analysis: AGCAAAAGCA/GCAAAAGCA/CAAAAGCA/AAAAGCA (7.776.959 reads). Each read starting exactly by these motifs 5′-AGCAAAAGCAGGGG/5′-GCAAAAGCAGGGG/5′-CAAAAGCAGGGG/5′-AAAAGCAGGGG were counted and plotted (Fig. 3H). Each read matching the exact sequence of a full-length replication product starting by 5′-AGC AAA...ACU-3′ (51-mer), 5′-GC AAA... ACU-3′ (50-mer), 5′-C AAA... ACU-3′ (49-mer) or 5′-_ AAA... ACU-3′ (48-mer) were counted and plotted as percentage (Fig. 3I).

## Electron microscopy

**Sample 1.** De novo pre-initiation and stalled elongation states were trapped by mixing 2.4 μM FluPol Zhejiang-H7N9 4 M with (i) 0.8 μM v51_mut_S, (ii) 8 μM chANP32A, (iii) 16 μM pS5 CTD(6mer) and (iv) 4 μM Anhui-H7N9 NP R416A in a final buffer containing 50 mM HEPES pH 8, 150 mM NaCl, 2 mM TCEP, 1 mM MgCl₂, ATP/GTP/CTP at 100 μM/NTP, and 100 μM of non-hydrolysable UpNHpp (Jena Bioscience).

De novo reaction mix was incubated for 4 h at 30 °C. The sample was centrifuged for 5 min at 11.000 g and kept at 4 °C before proceeding to grids freezing.

For grids preparation, 1.5 μl of sample was applied on each sides of plasma cleaned (Fischione 1070 Plasma Cleaner: 1 min 10 s, 90% oxygen, 10% argon) grids (UltrAufoil 1.2/1.3, Au 300). Excess solution was blotted for 3–5 s, blot force 0, 100% humidity, at 4 °C, with a Vitrobot Mark IV (ThermoFisher) before plunge-freezing in liquid ethane.

Automated data collection for sample 1 was performed on a TEM Titan Krios G3 (Thermo Fisher) operated at 300 kV equipped with a K3 (Gatan) direct electron detector camera and a BioQuantum energy filter, using EPU. Coma and astigmatism correction were performed on a carbon grid. Micrographs were recorded in counting mode at a ×105,000 magnification giving a pixel size of 0.84 Å with defocus ranging from −0.8 to −2.0 μm. Gain-normalised movies of 40 frames were collected with a total exposure of ~40 e⁻/Å².

**Sample 2.** The self-stalled elongation state was trapped using similar method as 'sample 1', but with UTP instead of the non-hydrolysable UpNHpp. The subsequent grid preparation and freezing steps are similar.

Automated data collection for sample 2 was performed on a TEM Glacios (Thermo Fisher) operated at 200 kV equipped with a F4i (Thermo Fisher) direct electron detector camera and a Selectris X energy filter, using EPU. Coma and astigmatism correction were performed on a carbon grid. Micrographs were recorded in counting mode at a ×130,000 magnification giving a pixel size of 0.878 Å with defocus ranging from −0.8 to −2.0 μm. EER movies were collected with a total exposure of ~40 e⁻/Å².

**Sample 3.** For Zhejiang-H7N9 PA K289A + C489R structures, 0.8 μM FluPol Zhejiang-H7N9 PA K289A + C489R was mixed with (i) 0.8 μM

v51_mut_S and 16 μM pS5 CTD(6mer) in a final buffer containing 50 mM HEPES pH 8, 150 mM NaCl, 2 mM TCEP. After 1 h incubation on ice, the sample was centrifuged for 5 min, 11.000 g and kept at 4 °C before proceeding to grids freezing.

For grid preparation, 1.5 μl of sample was applied on each sides of plasma cleaned (Fischione 1070 Plasma Cleaner: 1 min 10 s, 90% oxygen, 10% argon) grids (UltrAufoil 1.2/1.3, Au 300). Excess solution was blotted for 3–5 s, blot force 0, 100% humidity, at 4 °C, with a Vitrobot Mark IV (ThermoFisher) before plunge-freezing in liquid ethane.

Automated data collection was performed on a TEM Glacios (ThermoFisher) operated at 200 kV equipped with a F4i (Thermo-Fisher) direct electron detector camera and a Selectris X energy filter, using EPU. Coma and astigmatism correction were performed on a carbon grid. Micrographs were recorded in counting mode at a ×130,000 magnification giving a pixel size of 0.907 Å with defocus ranging from −0.8 to −2.0 μm. EER movies were collected with a total exposure of ~40 e⁻/Å².

## Image processing

For the TEM Titan Krios dataset (sample 1), movie drift correction was performed using Relion's Motioncor implementation, with 7 × 5 patch, using all movie frames[81]. All additional initial image processing steps were performed in cryoSPARC v3.3[82]. CTF parameters were determined using 'Patch CTF estimation', realigned micrographs were then manually inspected and low-quality images were manually discarded. To obtain an initial 3D reconstruction of FluPol Zhejiang-H7N9 4 M, particles were automatically picked on few hundreds micrographs using a circular blob with a diameter ranging from 100 to 140 Å. Particles were extracted using a box size of 360 × 360 pixels², 2D classified and subjected to an 'ab-initio reconstruction' job. The best initial model was further used to prepare 2D templates. Template picking was then performed using a particle diameter of 120 Å and particles extracted from dose-weighted micrographs. Successive 2D classifications were used to eliminate particles displaying poor structural features. All remaining particles were then transferred to Relion 4.0. Particles were divided in subset of 300k to 500k particles and subjected to multiple 3D classification with coarse image-alignment sampling using a circular mask of 180 Å. For each similar FluPol conformation, particles were grouped and subjected to 3D masked refinement followed by multiple 3D classification without alignment or using local angular searches. Once particles properly classified, bayesian polishing was performed and re-extracted 'shiny' particles were subjected to a last 3D masked refinement. Post-processing was performed in Relion 4.0 using an automatically or manually determined B-factor. For each final map, reported global resolution is based on the FSC 0.143 cut-off criteria. Local resolution variations were estimated in Relion 4.0. Detailed image processing information is shown in Fig. S7.

For the TEM Glacios datasets (sample 2 and 3), EER fractionation was set to 24, giving ~1e⁻/Å² per resulting fraction. Movie drift correction was performed using Relion's Motioncor implementation, with 5 × 5 patch[81]. All additional initial image processing steps were performed in cryoSPARC v4.0 or v4.3[82]. CTF parameters were determined using 'Patch CTF estimation', realigned micrographs were then manually inspected and low-quality images were manually discarded. To obtain an initial 3D reconstruction, particles were automatically picked on few hundreds micrographs using a circular blob with a diameter ranging from 100 to 140 Å. Particles were extracted using a box size of 380 × 380 pixels² (sample 2) or 340 × 340 pixels² (sample 3), 2D classified and subjected to an 'ab-initio reconstruction' job. The best initial model was further used to prepare 2D templates. Template picking was then performed using a particle diameter of 120 Å and particles extracted from dose-weighted micrographs. Successive 2D classifications were used to eliminate particles displaying poor structural features.

For sample 2, all remaining particles were subjected to a 'non-uniform refinement' job followed by a '3D classification' job. Particles

in a self-stalled elongation state were then subjected to a last non-uniform refinement. Post-processing was performed in Relion 4.0 using a manually determined B-factor. Reported global resolution is based on the FSC 0.143 cut-off criteria. Local resolution variations were estimated in Relion 4.0. Detailed image processing information is shown in Fig. S9.

For sample 3, all remaining particles were subjected to an 'Heterogeneous refinement' job, with three symmetrical dimers and three monomers as initial 3D models. Particles from well-resolved dimeric 3D classes were subjected to a final non-uniform refinement. Post-processing was performed in Relion 4.0 using an automatically determined B-factor. Reported global resolution is based on the FSC 0.143 cut-off criteria. Local resolution variations were estimated in Relion 4.0. Detailed image processing information is shown in Fig. S10.

### Model building and refinement

Using the FluPol Zhejiang-H7N9 elongation complex as starting point (PDB: 7QTL)[43], atomic models were constructed by iterative rounds of manual model building with COOT (version 0.9.8) and real-space refinement using Phenix (version 1.20.1)[83]. Validation was performed using Phenix (version 1.20.1). Model resolution according to the cryo-EM map was estimated at the 0.5 FSC cutoff. Figures were generated using ChimeraX (version 1.6.1)[84].

### Statistics

Statistics were performed with the GraphPad Prism software (Version 9.5.1 (528)), as indicated in the figure legends.

### Reporting summary

Further information on research design is available in the Nature Portfolio Reporting Summary linked to this article.

## Data availability

The coordinates and EM maps generated in this study have been deposited in the Protein Data Bank and the Electron Microscopy Data Bank: Infuenza A/Zhejiang-H7N9 polymerase (4 M mutant) in replicase-like conformation in pre-initiation state with RNAP II pS5 CTD peptide mimic bound in site 1A/2A, PDB 8PM0 and EMD-17755; Influenza A/Zhejiang-H7N9 polymerase (4 M mutant) in pre-initiation state with continuous RNAP II pS5 CTD peptide mimic bound in site 1A/2A PDB 8PNP and EMD-17782; Influenza A/Zhejiang-H7N9 polymerase (4 M mutant) in elongation state with continuous RNAP II pS5 CTD peptide mimic bound in site 1A/2A PDB 8PNQ and EMD-17783; Influenza A/Zhejiang-H7N9 polymerase (PA K289A + C489R) symmetric dimer bound to the promoter PDB 8POH and EMD-17792. Influenza A/H7N9 polymerase in pre-initiation state, intermediate conformation (I) with PB2-C(I), ENDO(T), and Pol II pS5 CTD peptide mimic bound in site 1A/2A PDB 8R3L and EMD-18872; Influenza A/H7N9 polymerase in self-stalled pre-termination state, with Pol II pS5 CTD peptide mimic bound in site 1A/2A PDB 8R3K and EMD-18871. The NGS raw reads generated in this study have been deposited in the European nucleotide archive under accession code ERP149587 [https://www.ebi.ac.uk/ena/browser/view/PRJEB64419]. The sequence files of plasmids used in this study have been deposited in the Zenodo repository [https://doi.org/10.5281/zenodo.10462746]. The raw data generated in this study are provided in the Source Data File. Source data are provided with this paper.

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

## Acknowledgements

We thank Ron Fouchier (Erasmus MC Department of Viroscience, Netherlands), George Brownlee (Oxford University), Bernard Delmas (INRAE, France), Robert-Jan Lebbink (Utrecht University, Netherlands) and Sandie Munier (Institut Pasteur, France) for sharing plasmids and antibodies. We thank Martin Pelosse for support in using the Eukaryotic Expression Facility at EMBL Grenoble; Aldo R. Camacho-Zarco and Martin Blackledge for providing huANP32A truncated constructs proteins; Daphne Walter and Vladimir Benes from the EMBL GeneCore facility for the next generation sequencing data; Wojtek Galej, Sarah Schneider and Romain Linares for access to the Glacios at EMBL Grenoble; Guy Schoehn and Eleftherios Zarkadas for access to the Glacios at IBS Grenoble; Aymeric Peuch for support with using the joint EMBL-IBS computer clusters. We acknowledge the European Synchrotron Radiation Facility and PSB for access to the Titan Krios CM01, and Michael Hons for assistance with data collection. We thank the Pasteur International Bioresources Network (PIBNet) for genome sequencing of recombinant viruses. This work was funded by the ANR grant FluTranscript (ANR-18-CE11-0028, S.C. and N.N.), the ANR LabEx IBEID (ANR-10-LABX-62-IBEID, N.N. and T.K.), and the German Research Foundation (DFG) grant SFB-TR84 (project B2, T.W.). This work used the platforms at the Grenoble Instruct-ERIC Center (ISBG; UMS 3518 CNRS CEA-UGA-EMBL) with support from the French Infrastructure for Integrated Structural Biology (FRISBI; ANR-10-INSB-05-02) and GRAL, a project of the University Grenoble Alpes graduate school (Ecoles Universitaires de Recherche) CBH-EUR-GS (ANR-17-EURE-0003) within the Grenoble Partnership for Structural Biology. The IBS Electron Microscope facility is supported by the Auvergne Rhône-Alpes Region, the Fonds Feder, the Fondation pour la Recherche Médicale and GIS-IBiSA. The funders had no role in study design, data collection and analysis, decision to publish, or preparation of the manuscript.

## Author contributions

TK, SC and NN conceived the project. TK and BA performed most of the experiments. CI performed serial passaging of mutant viruses. SP provided research assistance. MB and TW generated and provided essential reagents. TK, BA, CI, SC and NN oversaw and interpreted data. TK, BA, SC and NN wrote the manuscript with input from all authors. All authors read and approved the final manuscript.

## Competing interests

The authors declare no competing interests.
