## [Peer Review File · Nature Communications]

The host RNA polymerase II C-terminal domain is the anchor for replication of the influenza virus genomeREVIEWER COMMENTS

Reviewer #1 (Remarks to the Author):

Transcription and genome replication are two major processes important for influenza virus replication. These two processes depend on the same influenza polymerase (FluPol) but happen at different stages of the viral replication cycle. The machinery of controlling or switching between transcription and replication of the FluPol remains a long-standing and important research question. With advancement in structural and microscopic technology, increasing understanding and insights are being added to this area of study. In this manuscript, the authors propose a new role of FluPol and host RNA polymerase II binding in flu genome replication, in addition to transcription. The findings provide insight on the balance between flu genome transcription and replication. In general, the experimental design of the work is appropriate and the data obtained direct to the conclusion. There are a few minor comments that the authors may consider:

1. In figure 1D, the data was presented as mRNA : vRNA and cRNA : vRNA. However, supp fig 1E showed that vRNA level of different vRNP mutants were different, making it difficult to analyse the transcription activity and replication activity. In fact, both cRNA and vRNA levels increased in R638A mutant. Also, I wonder if there is any evidence to support the statement "it is likely that although a replication defect exists, it is masked by the transcription defect as more FluPols become available for replication activity..." (Lines 151-152).
2. For the serial passage experiment expressed in figure 2A and material and methods, MOI of 0.0001 was used between each passage. I would imagine less than 0.1% of the original was passed to the next passage. Will this introduce a bias to the input for passaging, with less frequent variant being unable to pass on?
3. The authors may want to elaborate on the choice of protein structure for analysis, particularly why a flu C polymerase was chosen. (Line 119).
4. In figure 3A-B, it is claimed that CTD-WT expression leads to a dose-dependent increase in FluPol-ANP32A binding and FluPol dimerization. The statistics of the bindings at various CTD expression levels are required (i.e. among the three dark grey bars).

Reviewer #2 (Remarks to the Author):

In the present study, Krischuns et al. investigate the role of the RNA polymerase II (RNAP II) C-terminal domain (CTD) in the replication of the influenza virus genome. The authors propose a model in which the host RNAP II is the anchor for transcription and replication of the viral genome, and provide evidence to support this model through functional and structural data, including:

- 1, Characterizing the role of the CTD-binding sites on the influenza virus polymerase (FluPol) in influenza virus genome replication. They demonstrate the correlation between FluPol activity and its association with CTD or ANP32 - a host factor of influenza virus which is essential for genome replication;
- 2, Showing enhanced de novo FluPol replication activity in the presence of CTD and ANP32 in an in vitro replication assay;
- 3, Presenting structural evidence that CTD binding to FluPol is compatible with its 'replicase' conformation.

In addition to these findings, there are other intriguing data, such as:

- 1, De novo replication at position 2 on the vRNA template in the in vitro assay;
- 2, Adaptive mutations arising from the passaging of a CTD-binding deficient mutant virus;
- 3, The role of these mutations in compensating for CTD binding, as determined through structural analysis.

Based on these findings, the authors assert that CTD binding to FluPol is not only crucial for genome transcription but also for replication, and propose a model for CTD-anchored transcription and replication of the influenza virus genome. Specifically, during genome replication, the CTD serves as a bridge between the influenza replication platform which comprises a resident FluPol in 'replicase' conformation and a free FluPol in the 'encapsidase' conformation, along with ANP32.

Overall, this study provides significant and novel insights into the mechanism of influenza virus genome replication, particularly highlighting the involvement of the host RNAP II CTD. These findings could potentially hold considerable implications for the development of new treatments or vaccines for influenza. However, further data are required to validate the proposed model. In addition, there is room for improvement in terms of the presentation. Specific points are provided below.

Specific points:

1, Regarding Fig. 1:

The correlations shown in Fig. 1F and 1G are challenging to comprehend. For instance, it remains unclear what level of the R value suggests a strong correlation. Is the correlation observed in Fig. 1G comparatively weaker due to its lower R value and a higher p value (lacking star significance)? In addition, the raw data linked with these panels (Fig. S1F and S1G) lack a positive control from the wildtype FluPol, a control that has been used in Fig. 1E. Also in these two panels the 'fold-change' values are calculated against the 'background', which has not been well defined.

Fig. 1C: absence of indication of statistical significance (stars are missing).

Fig. 1D and lines 145-146: the authors concluded that a decrease in mRNA:vRNA ratio and cRNA:vRNA ratio indicates both transcription and replication are affected, however, given that vRNA level is not constant in these experiments (see Fig. S1E), this conclusion which leans on the assumption that mRNA synthesis is proportional to vRNA levels, might not be true.

2, Regarding Fig. 3:

Fig. 3A: the 1.5x to 2x fold change in the split-luciferase assay appears to be quite subtle to draw a clear conclusion. Furthermore, is there statistical significance among the CTD-WT data points? This is important if the authors intend to claim a dose-dependent increase in ANP32A-binding.

Fig. 3B: I would assume such experiment indeed quantifies the level of symmetrical dimer of FluPol, which is not relevant to the context of this paper. The rationale presented in lines 232-237 are not convincing enough. This is particularly evident in Fig. S5C, only the data of FluPolB (which does not form the symmetrical dimer) looks significant enough. Moreover, Fig. S6C undoubtedly illustrates its measurement of the symmetrical dimer.

Fig. 3D: this panel shows a key piece of result to the paper, however, it lacks an in-depth elucidation or discussion. Notably, the authors annotated two bands as the 'Full-length' and 'Stalled' cRNA products, however, it is important to address the appearance of an additional band in lane 4 (also observed in Fig. 3E and Fig. S6) which is slightly exceeding the 40nt marker. The authors need to validate the sequence of these products by sequencing, in particular those bands slightly below and above the 40nt marker, to ensure the identity of the 34mer stalled product that they claimed. This would offer further insights into the role of CTD in regulating FluPol activity. Interestingly, in Fig. S5F, it seems there are many reads from products with lengths of 25-30nt and 30-35nt, compare with the 51mer full-length product, but these are not mentioned in the manuscript. In addition, the authors need to explain why there is a 'Stalled' cRNA product in lane 8, given that all four nucleotides were added into the sample.

3, Regarding Fig. 5:

Fig. 5J: the model clearly shows the clash of ANP32 and CTD when they bind to FluPol.

Additionally, I am curious about the authors' choice of using the H17N10 FluPol structure rather than the one presented in this study for the modelling of CTD. For instance, if the 8PNQ model, which features a continuous CTD, was used, the clash might become more pronounced.

Fig. 5K depicts the model of CTD's involvement in influenza virus genome transcription and replication proposed by the authors. Nevertheless, this model retains speculative due to the absence of substantial evidence, such as: a) structural evidence verifying that the CTD can bind to FluPol in 'encapsidase' conformation; b) structural or functional proof that the CTD and ANP32 can jointly bind to the FluPol; c) functional evidence to confirm whether the observations made in this study regarding CTD's facilitation of cRNA production from a vRNA template can be extended to vRNA production.

Furthermore, since only a small fragment of CTD (6x repeat) was used in the replication assay, extrapolating the conclusion drawing from the binding to these small peptides to the potential effects of binding a full-length peptide, or the complete host Pol II complex might not be entirely reasonable. It is important to consider that additional steric hindrance could be present in the latter scenarios. Further elaboration and discussion on this matter would be valuable.

Minor points:

- 1, Lines 170-172 and Fig. S1H, could the authors provide a discussion regarding why ANP32 binding is more severely affected in H7N9?
- 2, Fig. 2D: this panel should be put in Fig. S4.
- 3, Fig. 2F: the significance shown in this panel is confusing. The authors said 'Stars indicate statistical significance compared to the previous passage' but this makes no sense why there are stars on the data of RG - are these compared with the WT? Same issue for Fig. S3E.
- 4, Fig. 2G: this panel is quite messy and not clearly labelled. Only part of the blots is labelled but not all, there are some clashes in the middle panel, also the PA WT point in the right panel should be coloured black.
- 5, Fig. S3E, wrong labelling underneath the bar chart (PA K289A should be PA R454A)
- 6, Fig. S5A, error bars are missing for WT at later time points.
- 7, Fig. 3E, although the authors claim that in the presence of CTD, the ANP32 1-199 or 149-249 behave the same as the full-length, however, in the absence of CTD, the full-length ANP32 clearly made much more product than the truncated versions. This is worth mentioned.
- 8, The 4M mutant is introduced in lines 288-289, however it should be introduced earlier in line 273 (Fig. 3H/I & Fig. S5E).
- 9, Fig. 4A: calling this structure as a 'replicase-like' conformation is somewhat misleading. In all three structures shown in this figure, the PA endonuclease domain adopts a 'transcriptase' conformation, and the PB2-C is not visible. These structures reflect an 'intermediate' conformation between transcriptase and replicase.
- 10, Lines 683-686, could the authors specify the origin of the radiolabelled material, and verify the mentioned concentration? The amount of α -³²P ATP appears to be too little compared with the cold ATP used in the assay.
- 11, In the structure models, there are some phosphate groups missing on the Ser 5 residues. In model 8PNQ the density seems to be not good enough for modelling a continuous CTD peptide.

Reviewer #3 (Remarks to the Author):

The author provide compelling evidence of a role for RNAPII CTD in sustaining IAV viral replication. Cellular transcription is known to be essential for viral transcription, as IAV polymerase requires cap-snatched host RNA to prime transcription of the viral segments.

In this manuscript, the authors extend current knowledge of ANP32A (required for viral polymerase to function as a replicase) and provide strong evidence, both structural and biochemical, that RNAPII-ANP2 is also important for replication.

This and the accompanying work (focus on structural insights) provide a novel mechanism of how anchoring to the RNAPII CTD might allow viral polymerase to transcribe and replicate.

Lots of questions about viral RNA vs viral mRNA dynamics, role of host factors, consequence of cap-snatch vs replication for RNAPII remains, but I think this work is very solid and move the filed forward (despite never fully proving structurally the existence of RNAPII-ANP2-vPOL in replicase ode.).

Edits/clarifications that I think are needed before publication are below:

Can the author comment of how transcription vs replication are both coordinated by the RNAPII CTD but most of RNAPII enter in elongation mode after infection (as result of cap snatch)? Few papers have shown it very convincingly (Marazzi, Fodor, Benner labs). Is it possible that replication of Flu occurs not on promoter but when RNAPII is elongating? What is the phosphoform of RNAPII in replication vs cap snatch? I understand that some of these exp are elaborate and difficult/impossible to mimic in vitro, but a better explanation is needed for the model and interplay between what the author see in vitro and what it is seen on chromatin during infection. Essentially, the model disregard that RNAPII decrease during infection and moves away from promoters (increase in run through transcription). They need to reconcile their model with that. Lastly, that viral polymerase needs RNAPII in pausing/initiating stage (and that cap-snatching is cotranscriptional) has been shown not only by the authors, but also by Rialdi et al. in Cell the same year. It would be good to acknowledge that.

Response to reviewers of manuscript NCOMMS-23-35227-T

The host RNA polymerase II C-terminal domain is the anchor for replication of the influenza virus genome

We would like to thank the three reviewers for their constructive comments. In the revised version of the manuscript, additional data have been included and the text has been modified in order to address their concerns and suggestions.

A revised version of the manuscript with highlighted changes is provided.

Below is a point-by-point response to each of the reviewers comments.

Reviewer #1 (Remarks to the Author):

Transcription and genome replication are two major processes important for influenza virus replication. These two processes depend on the same influenza polymerase (FluPol) but happen at different stages of the viral replication cycle. The machinery of controlling or switching between transcription and replication of the FluPol remains a long-standing and important research question. With advancement in structural and microscopic technology, increasing understanding and insights are being added to this area of study. In this manuscript, the authors propose a new role of FluPol and host RNA polymerase II binding in flu genome replication, in addition to transcription. The findings provide insight on the balance between flu genome transcription and replication. In general, the experimental design of the work is appropriate and the data obtained direct to the conclusion.

There are a few minor comments that the authors may consider:

1. In figure 1D, the data was presented as mRNA : vRNA and cRNA : vRNA. However, supp fig 1E showed that vRNA level of different vRNP mutants were different, making it difficult to analyse the transcription activity and replication activity. In fact, both cRNA and vRNA levels increased in R638A mutant. Also, I wonder if there is any evidence to support the statement “it is likely that although a replication defect exists, it is masked by the transcription defect as more FluPols become available for replication activity...” (Lines 151-152).

We agree with the reviewer that Fig. S1E of the initial version of the manuscript shows different impacts on the mRNA, cRNA and vRNA accumulation levels for different vRNP mutations. To better highlight these differences in the revised version of the manuscript, we swapped and merged the three panels of Fig. S1E (now Fig. 1C) and the Fig. 1D (now Fig. S1E).

We state clearly that in the presence of the PA K289A and R638A mutants, cRNA and vRNA levels appear unchanged or even increased (page 7 lines 146-148 of the revised version).

In our assay the mRNA, cRNA and vRNA levels are interdependent, which makes it difficult to disentangle the v -> m, v-> c and c-> v activities of the polymerase, except when considering the mRNA:vRNA and cRNA:vRNA ratios. In the revised version of the manuscript, we removed the statement “it is likely that although a replication defect exists, it is masked by the transcription defect as more FluPols become available for replication activity...”, which was speculative and introducing an additional layer of complexity.

Instead, we merely stress the fact that like the R454A and 635A mutants, the PA K289A and R638A mutants show not only a ~80% decrease of mRNA:vRNA ratios bars but also a ~50% decrease of cRNA:vRNA ratios (Fig. S1E in the revised version), which points to an imbalance in the accumulation of replication products compared to the wild-type polymerase (page 7 lines 148-151).

2. For the serial passage experiment expressed in figure 2A and material and methods, MOI of 0.0001 was used between each passage. I would imagine less than 0.1% of the original was passed to the next passage. Will this introduce a bias to the input for passaging, with less frequent variant being unable to pass on?

The indication of an MOI of 0.0001 was not entirely accurate, we apologize for that.

In order to achieve the right balance between a too high input (which favours the accumulation of defective particles at the expense of infectious ones) and a too low input (which introduces a bias against less frequent variants), three dilutions of passage N (1:10, 1:100 and 1:1000) were used to pass to passage N+1. The supernatant corresponding to one of these dilutions (the lowest dilution that resulted in the infectious titers greater than or equal to titers at passage N) was selected to pass on passage N+2. These conditions resulted in MOIs in the 0.001 or 0.0001 range, depending on the passages.

In the revised version, this procedure is described in the Material and Methods section (page 25 lines 617-621), and we comment on the fact that less frequent variants may have been lost in these conditions, and that the reversion pathways we observe only represent a fraction of possible reversion pathways (page 9 lines 189-191).

3. The authors may want to elaborate on the choice of protein structure for analysis, particularly why a flu C polymerase was chosen. (Line 119).

To date, there is no structure published for the Flu_APol replicase-ANP32-encapsidase complex. Therefore, we have built a model for Flu_APol based on the Flu_CPol replicase-ANP32-encapsidase complex by superposing equivalent Flu_APol domains. We clarified this point as follows: “As there are no structures published for an equivalent Flu_APol complex, we used the Flu_CPol structures to generate a model of the Flu_APol replicase-encapsidase complex” (page 6 lines 111-112 of the revised version).

4. In figure 3A-B, it is claimed that CTD-WT expression leads to a dose-dependent increase in FluPol-ANP32A binding and FluPol dimerization. The statistics of the bindings at various CTD expression levels are required (i.e. among the three dark grey bars)

We agree with the reviewer that appropriate statistics are needed to claim dose-dependency. They are shown below.

(A) In the FluPol ANP32A-binding assay (A), the fold-changes are significantly different between the highest and the lowest CTD-WT concentration, while they are not significant for the CTD-S5A control.

(B) In the FluPol dimerisation assay (which we renamed as “FluPol oligomerisation” throughout the manuscript in view of comment #5 of reviewer 2: please see our detailed response below), the fold-changes are significantly different between the high, middle and low CTD-WT concentrations. Some level of significance is also observed with the CTD-S5A control, which we believe is misleading. Indeed, when looking at the raw luciferase reading values (C), it appears that the dose-dependent increase in the mCherry-CTD-S5A/mCherry ratio is mainly due to a decrease in the luciferase signal with increasing doses of mCherry-NLS, rather than an increase in the luciferase signal with increasing doses of mCherry-CTD-S5A.

However, we agree that dose-dependency is not formally demonstrated and cannot be claimed. We rephrased the text accordingly (page 10 lines 222-226 of the revised version).

Reviewer #2 (Remarks to the Author):

In the present study, Krischuns et al. investigate the role of the RNA polymerase II (RNAP II) C-terminal domain (CTD) in the replication of the influenza virus genome. The authors propose a model in which the host RNAP II is the anchor for transcription and replication of the viral genome, and provide evidence to support this model through functional and structural data, including:

- 1, Characterizing the role of the CTD-binding sites on the influenza virus polymerase (FluPol) in influenza virus genome replication. They demonstrate the correlation between FluPol activity and its association with CTD or ANP32 - a host factor of influenza virus which is essential for genome replication;
- 2, Showing enhanced de novo FluPol replication activity in the presence of CTD and ANP32 in an in vitro replication assay;
- 3, Presenting structural evidence that CTD binding to FluPol is compatible with its 'replicase' conformation.

In addition to these findings, there are other intriguing data, such as:

- 1, De novo replication at position 2 on the vRNA template in the in vitro assay;
- 2, Adaptive mutations arising from the passaging of a CTD-binding deficient mutant virus;
- 3, The role of these mutations in compensating for CTD binding, as determined through structural analysis.

Based on these findings, the authors assert that CTD binding to FluPol is not only crucial for genome transcription but also for replication, and propose a model for CTD-anchored transcription and replication of the influenza virus genome. Specifically, during genome replication, the CTD serves as a bridge between the influenza replication platform which comprises a resident FluPol in 'replicase' conformation and a free FluPol in the 'encapsidase' conformation, along with ANP32.

Overall, this study provides significant and novel insights into the mechanism of influenza virus genome replication, particularly highlighting the involvement of the host RNAP II CTD. These findings could potentially hold considerable implications for the development of new treatments or vaccines for influenza. However, further data are required to validate the proposed model. In addition, there is room for improvement in terms of the presentation. Specific points are provided below.

Specific points:

- 1, Regarding Fig. 1:

The correlations shown in Fig. 1F and 1G are challenging to comprehend. For instance, it remains unclear what level of the R value suggests a strong correlation. Is the correlation observed in Fig. 1G comparatively weaker due to its lower R value and a higher p value (lacking star significance)? According to Mukaka et al, 2012 (PMID: 23638278), correlation coefficients between 0.9-1 indicate a very high positive correlation, 0.7-0.9 high positive correlation and 0.5-0.7 moderate positive correlation. The p values indicate the likelihood that such a correlation is occurring by chance. We believe the lack of significance observed for the replication products ($p > 0.05$) is due to low sample size and high variation in the qRT-PCR quantification of vRNAs and cRNAs. However, given the high p value in Fig. 1G of the initial version of the manuscript, we decided not to show the correlation graphs. Instead, in Fig. 1E-F of the revised version of the manuscript, we show the fold-changes of trans-complementation compared to background (Fig. S1F-G in the initial version). The text (page 7-8 lines 158-162) was modified accordingly.

In addition, the raw data linked with these panels (Fig. S1F and S1G) lack a positive control from the wildtype FluPol, a control that has been used in Fig. 1E.

Also in these two panels the 'fold-change' values are calculated against the 'background', which has not been well defined.

We thank the reviewer for pointing this out.

The trans-complementation assays (Fig. S1F-G in the initial version, switched to the main Fig. 1E-F in the revised version) were repeated in the presence of the controls shown in Fig. 1D.

The data are represented as the fold-change increase in FluPol activity signal when the FluPol_(CTD-) is co-transfected with the FluPol_(R-) or FluPol_(T-), over the background defined as "the sum of the signals measured when the FluPol_(CTD-) or the FluPol_(R-/T-) were transfected alone" (page 7 lines 156-157 and page 41 lines 1127-1130). The FluPol_(R-) + FluPol_(T-) data are shown on the left of the graphs in Fig. 1E-F to provide a positive control for trans-complementation.

Fig. 1C: absence of indication of statistical significance (stars are missing).

Indications of significance got lost in the final figure preparation and have been added. We apologise for this oversight.

Fig. 1D and lines 145-146: the authors concluded that a decrease in mRNA:vRNA ratio and cRNA:vRNA ratio indicates both transcription and replication are affected, however, given that vRNA level is not constant in these experiments (see Fig. S1E), this conclusion which leans on the assumption that mRNA synthesis is proportional to vRNA levels, might not be true.

We agree with the reviewer that Fig. S1E of the initial version of the manuscript shows different impacts on vRNA accumulation levels for different vRNP mutations. To better highlight these differences in the revised version of the manuscript, we swapped and merged the three panels of Fig. S1E (now Fig. 1C) and the Fig. 1D (now Fig. S1E).

In our assay the mRNA, cRNA and vRNA levels are interdependent, which makes it difficult to disentangle the v → m, v → c and c → v activities of the polymerase, except when considering the mRNA:vRNA and cRNA:vRNA ratios. We did not mean to assert that mRNA/cRNA synthesis is strictly proportional to vRNA levels, therefore we rephrased our conclusion as follows: "like the R454A and 635A mutants, the PA K289A and R638A mutants show not only a ~80% decrease of mRNA:vRNA ratios bars but also a ~50% decrease of cRNA:vRNA ratios (Fig. S1E in the the revised version), indicating an imbalance in the accumulation of replication products compared to the WT polymerase" (page 7 lines 148-151 of the revised version).

2, Regarding Fig. 3:

Fig. 3A: the 1.5x to 2x fold change in the split-luciferase assay appears to be quite subtle to draw a clear conclusion. Furthermore, is there statistical significance among the CTD-WT data points? This is important if the authors intend to claim a dose-dependent increase in ANP32A-binding.

We agree with the reviewer that appropriate statistics are necessary to claim dose-dependency.

They are shown below.

The fold-changes are significantly different between the highest and the lowest CTD-WT concentration, while they are not significant for the CTD-S5A control. However, also in view of

comment #4 of reviewer 1 (please see our detailed response above), we decided to remove our statement about dose-dependency and have rephrased the text accordingly (page 10 lines 222-226 of the revised version).

Fig. 3B: I would assume such experiment indeed quantifies the level of symmetrical dimer of FluPol, which is not relevant to the context of this paper. The rationale presented in lines 232-237 are not convincing enough. This is particularly evident in Fig. S5C, only the data of FluPolB (which does not form the symmetrical dimer) looks significant enough. Moreover, Fig. S6C undoubtedly illustrates its measurement of the symmetrical dimer.

To control whether the signal in Fig. 3B is representing a measurement of the symmetrical dimer we repeated the assay in the presence of the PB2 NEQ71-73AAA mutant, which was previously shown to suppress symmetrical FluPol dimerisation. In the presence of the PB2 mutant, overexpression of the CTD still results in an increased FluPol-FluPol interaction signal, indicating that FluPol oligomers distinct from the symmetrical dimers are involved.

These new data are presented in Fig. S5C-E of the revised version of the manuscript, and the issue about which type of FluPol oligomer is involved is discussed more thoroughly (page 10 lines 230-240). We state that it is unclear which oligomeric species are affected by CTD overexpression, and we systematically replaced “FluPol dimerisation” by “FluPol oligomerisation” when we refer to the signal measured in our split-luciferase assay.

Fig. 3D: this panel shows a key piece of result to the paper, however, it lacks an in-depth elucidation or discussion. Notably, the authors annotated two bands as the ‘Full-length’ and ‘Stalled’ cRNA products, however, it is important to address the appearance of an additional band in lane 4 (also observed in Fig. 3E and Fig. S6) which is slightly exceeding the 40nt marker. The authors need to validate the sequence of these products by sequencing, in particular those bands slightly below and above the 40nt marker, to ensure the identity of the 34mer stalled product that they claimed. This would offer further insights into the role of CTD in regulating FluPol activity. Interestingly, in Fig. S5F, it seems there are many reads from products with lengths of 25-30nt and 30-35nt, compare with the 51mer full-length product, but these are not mentioned in the manuscript. In addition, the authors need to explain why there is a ‘Stalled’ cRNA product in lane 8, given that all four nucleotides were added into the sample.

The referee is correct that discussion of the gels is inadequate and we have therefore adapted the text (page 11-12 lines 261-278). There are indeed three major products formed, which run at approximately 38-40, 45 and 55-56 nts. The 45 nts product is only seen with ATP, CTP and GTP present and we have difficulty explaining the origin of this product. The ~55 nts product is strong with all NTPs and very weak with just ATP, CTP and GTP. With only ATP, CTP and GTP, RNA synthesis is expected to stall at A18 of the template just before the 5' hook (without U incorporation, or with UpNHpp in the elongation structure) yielding a product of 33 or 32 nts, with or without incorporation of A1. With all NTPs, the full-length product with hook readthrough is 51 nts or 50 nts, if A1 is not incorporated, as suggested by NGS sequencing data. That there is a stalled product even when all NTPs are present shows that the tightly bound hook still presents a barrier to RNA synthesis under these *in vitro* conditions. Indeed, we now include a new cryoEM structure (Fig. S9 and Table 1b in the revised version) that shows that this stalled product is in fact a 35-mer (or 34-mer if A1 is not incorporated) since template U17 is observed to base-pair with A35 (or A34) of the product at the +1 position (it is a product complex with PPI). This structure complements the already included elongation complex with ATP, GTP, CTP, UpNHpp which has a 33/32 nts product.

The difference between the theoretical stalled and full-length products is 16-18 nts, which corresponds to the difference between the two major bands observed with all NTPs present, even if these bands are 5-6 nts longer than predicted, according to the ladder. This was the rationale of

labelling them as full-length cRNA product and stalled cRNA product, but we agree this has not been formally shown, so now we are now more circumspect in the text and annotate them more tentatively with a '?'. We also note that on the full gels (Fig. S6A-B in the revised version), abortive products are observed throughout the length range, but notably strong in the 20-30 nts region, consistent with the sequencing data.

Even though there are uncertainties in the assignment of the products, the main purpose of this figure is to show the synergistic effect of CTD and ANP32 to significantly enhance *de novo* product formation. Furthermore, this seems to be primarily dependent on the LCAR of ANP32.

3, Regarding Fig. 5:

Fig. 5J: the model clearly shows the clash of ANP32 and CTD when they bind to FluPol. Additionally, I am curious about the authors' choice of using the H17N10 FluPol structure rather than the one presented in this study for the modelling of CTD. For instance, if the 8PNQ model, which features a continuous CTD, was used, the clash might become more pronounced.

Fig. 5K depicts the model of CTD's involvement in influenza virus genome transcription and replication proposed by the authors. Nevertheless, this model retains speculative due to the absence of substantial evidence, such as: a) structural evidence verifying that the CTD can bind to FluPol in 'encapsidase' conformation; b) structural or functional proof that the CTD and ANP32 can jointly bind to the FluPol;

We agree with the reviewer that there may well be a clash between ANP32 and CTD when bound to FluPol_(E). In the revised version of the manuscript, we removed all reference to CTD simultaneously binding at the interface between FluPol_(E) and ANP32, which was very speculative. We concentrate now on the finding that the FluPol_(R) can be bound by the CTD, and CTD, together with ANP32, enhances replication activity. In this light, Fig 1A in the initial version of the manuscript was moved (now Fig. S1A), the former Fig. S1A was removed and Fig. 1A was adapted to illustrate the location of FluPol CTD-binding residues in both the FluPol_(E) and FluPol_(R) in the model of the Flu_APol replication complex. Moreover, Fig. 5J and Fig. S10K-L from the initial version were removed, the model (Fig. 5K in the initial version) was modified accordingly and is now represented in a separate figure (Fig. 6 in revised version), and the text was updated accordingly (page 17 lines 426-430).

c) Functional evidence to confirm whether the observations made in this study regarding CTD's facilitation of cRNA production from a vRNA template can be extended to vRNA production. The model as drawn in Fig. 6 and described in the legend of the figure is limited to the switch between FluPol transcription and FluPol replication (vRNA → cRNA synthesis). Although we agree with the reviewer that an extension of the model to cRNA → vRNA synthesis is speculative, we feel it is interesting to mention this possibility in the discussion and to stress the fact that further studies are needed to explore functional differences between vRNP and cRNP genome replication activities (page 19 lines 466-472 in the revised version).

Furthermore, since only a small fragment of CTD (6x repeat) was used in the replication assay, extrapolating the conclusion drawing from the binding to these small peptides to the potential effects of binding a full-length peptide, or the complete host Pol II complex might not be entirely reasonable. It is important to consider that additional steric hindrance could be present in the latter scenarios. Further elaboration and discussion on this matter would be valuable.

Since we now only consider canonical CTD binding to site 1A and 2A for the replicase, this is not so different to binding to the transcriptase, where steric hinderance is not thought to be a problem. The continuous density between the two sites implicates a total of 5 repeats, for which the peptide used is sufficient. What happens in the context of the whole RNAP II complex with full-length CTD we don't currently know and this point needs further investigation, as stated in the discussion

section, page 20 lines 493-496 of the revised version. We don't think that CTD continuity between binding in sites 1A and 2A is a necessity. We note that replicating FluPol has no need to access the RNAP II RNA exit channel and we imagine the CTD is acting as a platform to facilitate assembly of the components needed for replication.

Minor points:

1, Lines 170-172 and Fig. S1H, could the authors provide a discussion regarding why ANP32 binding is more severely affected in H7N9?

Based on the sequence alignments (Fig. S2) and available structural information, one would expect the interaction of Anhui-H7N9 and WSN-FluPol to ANP32 to be similar, which is supported by the fact that the trends observed for the investigated PA mutants are indeed very much conserved between WSN (Fig. 1B) and Anhui-H7N9 (Fig. S1F). To ensure that differences are not due to differential expression levels, we have now included a western blot of the tested Anhui-H7N9 mutants (Fig. S1G). For us, the most likely explanation for the overall observed stronger phenotypes on huANP32A-binding in the Anhui-H7N9 background is due to the fact that when binding to huANP32A is investigated in the split-gaussia luciferase assay, Anhui-H7N9 shows an increased dynamic range compared to the WSN FluPol which leads to seemingly increased phenotypes of the investigated mutants. However, the trends stay very much comparable. Consistent with this hypothesis, when chANP32A is used instead of huANP32A (dark blue bars added in the Fig. 1B and Fig. S1F in the revised version), the binding assay of Anhui-H7N9 and WSN to chANP32 shows a similar dynamic range, which goes along with similar binding defects of WSN and Anhui-H7N9.

2, Fig. 2D: this panel should be put in Fig. S4.

We thank the reviewer for this suggestion. We feel that it is helpful to show the reader once the position of second-site mutations on a 3D FluPol structure in the main figures. Moreover, we have now used a colour-code to help the reader assign a given second-site mutation to a primary CTD-binding mutation in Fig. 2D as well as in Fig. 2G (see below).

3, Fig. 2F: the significance shown in this panel is confusing. The authors said 'Stars indicate statistical significance compared to the previous passage' but this makes no sense why there are stars on the data of RG - are these compared with the WT? Same issue for Fig. S3E.

We thank the reviewer for pointing this out. We have added the relevant information to the legends of Fig. 2F and S3E (page 42, lines 1167-1169 of the revised version and in the legend of Fig. S3E), as follows: "Stars indicate statistical significance when passage N is compared to passage N-1, or in the case of the RG, statistical significance is indicated compared to WT FluPol."

4, Fig. 2G: this panel is quite messy and not clearly labelled. Only part of the blots is labelled but not all, there are some clashes in the middle panel, also the PA WT point in the right panel should be coloured black.

We apologise for the mislabelling of the PA WT data point in the FluPol DDX5-binding panel and we corrected it. We agree with the reviewer that Fig. 2G is a complex figure as it merges the four reversion pathways with respect to their impact on four variables (FluPol activity, huANP32A-binding, CTD-binding and DDX5-binding).

We improved the graphs shape and labelling in the middle panel to limit clashes as much as possible. We tried to label every single combination of primary and secondary FluPol mutations, unfortunately this made the graphs harder to read than when only the starting points (K289A, R454A, K635A, R638A) and the WT reference were labelled. As a compromise, we used the same colour-code as in Fig. 2D to help the reader identify each distinct reversion pathways in these graphs. Moreover, Fig. 2D now refers to these colours. Although not every single combination of primary and second-site mutation is indicated, individual values are shown in Fig. 2F and Fig. S3E, as indicated in the legend of Fig. 2G, and are thereby available for the reader.

In our opinion, it is meaningful to provide such an overview of the changes observed upon serial passaging of CTD-binding mutant viruses, to convey the message of a trend towards increasing levels of huANP32A binding as well as CTD-binding. However, we are willing to switch Fig. 2G to the supplementary Figures if the reviewer feels that it is preferable.

5, Fig. S3E, wrong labelling underneath the bar chart (PA K289A should be PA R454A)
We thank the reviewer for pointing this out and have corrected the wrong labelling.

6, Fig. S5A, error bars are missing for WT at later time points.
The “CTD-WT at 108 hpt” condition represents the 100% reference to which the other data points are referred to, which is the reason why it shows no error bars. As shown below, the signals measured at 96 and 84 hpt are very homogeneous and therefore the SD is very small.

X	Group A						Group B						Group C					
	pCI-empty						pCI mCherry CTD-SV40NLS						pCI-mCherry-CTD-SV40NLS-S5A					
Hours	A:Y1	A:Y2	A:Y3	A:Y4	A:Y5	A:Y6	B:Y1	B:Y2	B:Y3	B:Y4	B:Y5	B:Y6	C:Y1	C:Y2	C:Y3	C:Y4	C:Y5	C:Y6
0.00000000	0.001782	0.000000	0.000000	0.000000	0.000000	0.000000	0.000000	0.000000	0.000000	0.000000	0.000000	0.000000	0.000000	0.000000	0.000000	0.000000	0.000000	0.000000
12.00000000	0.001769	0.000000	0.000000	0.000000	0.000000	0.000000	0.001315	0.000906	0.000000	0.011383	0.342310	0.164920	0.000984	0.000315	0.001215	0.015509	0.175170	0.068880
24.00000000	0.002051	0.000000	0.000000	0.000000	0.000000	0.000000	1.445820	1.338133	1.218380	1.549232	7.226910	5.482080	1.856099	1.268591	1.446410	1.722724	5.098850	3.488840
36.00000000	0.001965	0.000000	0.000000	0.000000	0.000000	0.000000	6.054223	6.445921	5.416873	5.874272	14.28290	10.70520	7.488323	6.157153	6.133384	6.653600	10.69060	7.617640
48.00000000	0.002219	0.000000	0.000000	0.000000	0.000000	0.000000	18.60909	17.97394	15.21154	17.88195	23.76670	19.26440	23.25093	19.28400	18.19383	19.35365	20.13540	15.95570
60.00000000	0.001911	0.000000	0.000000	0.000000	0.000000	0.000000	48.91250	47.90866	42.46583	45.80324	38.33760	33.56730	60.69000	54.03749	50.76545	53.48193	38.93440	32.79870
72.00000000	0.002219	0.000000	0.000000	0.000000	0.000000	0.000000	81.08652	81.67359	75.52854	77.16348	73.44650	68.43240	95.85894	91.38933	85.44712	84.15107	83.78790	64.80550
84.00000000	0.001984	0.000000	0.000000	0.000000	0.000000	0.000000	94.24323	95.60055	92.85753	92.96801	93.92420	89.62940	113.8392	106.8588	101.3745	97.01699	110.2820	89.18100
96.00000000	0.002096	0.000000	0.000000	0.000000	0.000000	0.000000	98.96825	99.45657	98.31583	98.28469	102.2260	101.5340	120.6465	113.1611	107.7303	102.8119	130.4090	112.1060
108.00000000	0.002713	0.000000	0.000000	0.000000	0.000000	0.000000	100.0000	100.0000	100.0000	100.0000	100.0000	100.0000	122.5623	115.6593	110.7968	105.0782	132.3850	115.0870

7, Fig. 3E, although the authors claim that in the presence of CTD, the ANP32 1-199 or 149-249 behave the same as the full-length, however, in the absence of CTD, the full-length ANP32 clearly made much more product than the truncated versions. This is worth mentioned.

We agree with the reviewer’s comment and have modified the text accordingly with the following statement (page 12 lines 283-286): “While without addition of CTD, only FL huANP32A supported significant replication product synthesis, we show that the 144-199 region of huANP32A is most critical to observe the synergistic CTD- and ANP32-dependent enhancement of replication product synthesis”

8, The 4M mutant is introduced in lines 288-279, however it should be introduced earlier in line 273 (Fig. 3H/I & Fig. S5E).

We thank the reviewer for pointing out this mistake of ours and have moved the description of the 4M mutant (page 12 lines 292-295) in the revised version, so that it is introduced when it is first mentioned.

9, Fig. 4A: calling this structure as a ‘replicase-like’ conformation is somewhat misleading. In all three structures shown in this figure, the PA endonuclease domain adopts a ‘transcriptase’ conformation, and the PB2-C is not visible. These structures reflect an ‘intermediate’ conformation between transcriptase and replicase.

We used the term ‘replicase-like’ to indicate that it is not the full replicase conformation; it does have the characteristic replicase position of the mid-link and cap-binding domain, now denoted PB2-C(R), even if the endonuclease is transcriptase-like, now denoted ENDO(T). However, we agree with the referee that the discussion needs to be more precise. In fact, we have observed several different conformations of the Zhejiang-H7N9 4M polymerase performing *de novo* RNA synthesis in this data collection and in subsequent ones. These include

- Initiation with Endo(T), PB2-C (R) (Fig. 4A, middle panel)
- Initiation with Endo(T), no PB2-C (Fig. 4B)
- Initiation with novel intermediate structure first described by Keown et al. (PMID: 35017564, PDB:7NHX). This is in fact a structure initially named ‘transcriptase-like’ (Fig. S7 in red of the

initial version, not treated further), which we have now analysed fully and included in this manuscript (Fig. 4A left panel, Table 1a, Fig. S7, page 13-14 lines 322-328 in the revised version). This structure, observed in the context of *de novo* RNA synthesis supports the idea that this is a true intermediate state between transcriptase and replicase as proposed by Li et al, PMID: 37488357, PDB:8H69).

- Elongation stalled with UpNHpp, with Endo(T) (Fig. 4E).
- Elongation self-stalled at U17, with Endo(T). This is a new structure, now included in the manuscript, which shows that even when all NTPs are present self-stalling prior to the hook occurs (Fig. S9, Table 1b, page 14 lines 335-342 in the revised version).

We already stated ‘Altogether, these results show that *in vitro*, *de novo* RNA synthesis assays result in a heterogeneous mix of FluPol conformations including replicase-like and transcriptase-like configurations, reflecting the flexibility of the peripheral PB2-C domains.’

We believe this a fair statement, although, based on our further analysis, we have updated it to (page 14-15 lines 348-367 in the revised version):

‘Altogether, these results show that *in vitro*, *de novo* RNA synthesis assays result in a heterogeneous mix of FluPol conformations intermediate between the full replicase and transcriptase configurations, reflecting the flexibility of the peripheral PB2-C and PA-N domains (Fig. S7). Importantly, a replicase-like initiation complex (PB2-C(R)/ENDO(T)) is observed for the first time, characterised by the radically different position of the cap-binding domain compared to the transcriptase conformation (Fig. 4A) (Thierry et al., 2016), which was never observed during extensive studies of cap-dependent transcription (Wandzik et al., 2020). However, the observed stalled elongation state, in which the translocated 3' end of the template has reached the secondary binding site (Fig. 4E), is similar to the previously described pre-termination transcription state since PB2-C is not visible (Wandzik et al., 2020). Finally, all structures have the CTD peptide mimic bound in both sites 1A and 2A, suggesting that both replication and transcription are consistent with CTD binding (Fig. 4A-B, E). The observation of the intermediate state (PB2-C(I)/ENDO(T)) in the context of *de novo* RNA synthesis lends support to the suggestion that it is an important intermediate in the transition from transcriptase to replicase (Li et al, PMID: 37488357). Indeed, our structural results suggest that this transition might occur in the following sequence of steps: PB2-C(T)/ENDO(T), PB2-C(I)/ENDO(T), PB2-C(R)/ENDO(T), PB2-C(R)/ENDO(R), with the first and last being the full transcriptase and replicase, respectively. We note that the final step, ENDO(T) to ENDO(R) can only happen after PB2-C(R) is established, thus rotating the PB2-NLS domain into a position where it can interact with ENDO(R). Furthermore, we suggest that the full replicase conformation (including the otherwise flexible 627-domain) is only stabilised in the context of the replicase-encapsidase complex.’

10, Lines 683-686, could the authors specify the origin of the radiolabelled material, and verify the mentioned concentration? The amount of α -32P ATP appears to be too little compared with the cold ATP used in the assay.

The reviewer is correct. The concentration of α -32P ATP has been corrected and the origin of the radiolabelled material has been added in the “Material and Methods” section (page 29 lines 727-729 in the revised version).

11, In the structure models, there are some phosphate groups missing on the Ser 5 residues. In model 8PNQ the density seems to be not good enough for modelling a continuous CTD peptide. The density for the linker between CTD sites 1A and 2A is not as strong as the density in the binding sites. It is best visible in unsharpened maps. Even so there is no clear density for the phosphorylated serine in the linker, so the phosphate was omitted. In the unsharpened map for 8PNQ, there is clear density linking the two sites and the model was taken over from structures with better defined maps.

Reviewer #3 (Remarks to the Author):

The author provide compelling evidence of a role for RNAPII CTD in sustaining IAV viral replication. Cellular transcription is known to be essential for viral transcription, as IAV polymerase requires cap-snatched host RNA to prime transcription of the viral segments.

In this manuscript, the authors extend current knowledge of ANP32A (required for viral polymerase to function as a replicase) and provide strong evidence, both structural and biochemical, that RNAPII-ANP2 is also important for replication.

We thank the reviewer for his assessment of our manuscript and have addressed his critics below.

This and the accompanying work (focus on structural insights) provide a novel mechanism of how anchoring to the RNAPII CTD might allow viral polymerase to transcribe and replicate.

Lots of questions about viral RNA vs viral mRNA dynamics, role of host factors, consequence of cap-snatch vs replication for RNAPII remains, but I think this work is very solid and move the filed forward (despite never fully proving structurally the existence of RNAPII-ANP2-vPOL in replicase ode.).

Edits/clarifications that I think are needed before publication are below:

Can the author comment of how transcription vs replication are both coordinated by the RNAPII CTD but most of RNAPII enter in elongation mode after infection (as result of cap snatch)? Few papers have shown it very convincingly (Marazzi, Fodor, Benner labs). Is it possible that replication of Flu occurs not on promoter but when RNAPII is elongating? What is the phosphoform of RNAPII in replication vs cap snatch? I understand that some of these exp are elaborate and difficult/impossible to mimic in vitro, but a better explanation is needed for the model and interplay between what the author see in vitro and what it is seen on chromatin during infection.

Essentially, the model disregard that RNAPII decrease during infection and moves away from promoters (increase in run through transcription). They need to reconcile their model with that.

We agree with the reviewer that in the previous version of the manuscript, how FluPol functions in the context of on-going host RNAP II transcription and influenza virus-induced RNAP II dysregulation was not properly discussed.

In the revised version of the manuscript, we discuss the question how the CTD is coordinating transcription versus replication in the context of chromatin in infected cells, and we refer to the existing evidence for dysregulation of RNPA II function in infected cells (page 19-20 lines 474-497).

Lastly, that viral polymerase needs RNAPII in pausing/initiating stage (and that cap-snatching is cotranscriptional) has been shown not only by the authors, but also by Rialdi et al. in Cell the same year. It would be good to acknowledge that.

We have added the Rialdi et al. to the manuscript (Ref #24) which is now cited both in the introduction (page 4 line 75) and in the discussion (page 19 line 486) of the revised version.

REVIEWERS' COMMENTS

Reviewer #1 (Remarks to the Author):

The authors have satisfactorily addressed the comments and revised the manuscript accordingly. I have no further comments on the revised version.

Reviewer #2 (Remarks to the Author):

In the revised manuscript, the authors have successfully addressed most of my concerns. The quality of the manuscript has been significantly simplified and improved, and the conclusions now appear clearer and more accurate to me. I have no further concerns at the moment.

Reviewer #4 (Remarks to the Author):

The author answered all the points raised. I support the publication of this manuscript.

Response to reviewers of manuscript NCOMMS-23-35227A

The host RNA polymerase II C-terminal domain is the anchor for replication of the influenza virus genome

We would like to thank again all reviewers for their constructive comments and for helping us improve our manuscript.

Reviewer #1 (Remarks to the Author):

The authors have satisfactorily addressed the comments and revised the manuscript accordingly. I have no further comments on the revised version.

Reviewer #2 (Remarks to the Author):

In the revised manuscript, the authors have successfully addressed most of my concerns. The quality of the manuscript has been significantly simplified and improved, and the conclusions now appear clearer and more accurate to me. I have no further concerns at the moment.

Reviewer #4 (Remarks to the Author):

The author answered all the points raised. I support the publication of this manuscript.